# *Periplaneta americana* Extract Protects Glutamate-Induced Nerve Cell Damage by Inhibiting *N-Methyl-D-Aspartate* Receptor Activation

**DOI:** 10.3390/biology14020193

**Published:** 2025-02-13

**Authors:** Yongfang Zhou, Xin Yang, Canhui Hong, Tangfei Guan, Peiyun Xiao, Yongshou Yang, Chenggui Zhang, Zhengchun He

**Affiliations:** 1Yunnan Provincial Key Laboratory of Entomological Biopharmaceutical R&D, College of Pharmacy, Dali University, Dali 671000, China; zhouyongfang0706@163.com (Y.Z.); 17387260406@163.com (X.Y.); hongch09@163.com (C.H.); guantangfei2021026@163.com (T.G.); xiaopeiyun@dali.edu.cn (P.X.); yangyongshou@dali.edu.cn (Y.Y.); chenggui_zcg@163.com (C.Z.); 2National-Local Joint Engineering Research Center of Entomoceutics, Dali 671000, China

**Keywords:** *Periplaneta americana* (L.), network pharmacology, oxidative stress, excitatory neurotoxicity, NMDAR

## Abstract

Neurodegenerative diseases are a global health concern. This study expands on prior findings that *Periplaneta americana* (L.) extract (PAS840) protects PC12 cells from H_2_O_2_-induced injury. Using Alzheimer’s disease as an example, we employed network pharmacology and molecular docking to analyze PAS840’s substance basis and predict target effects. In vitro experiments were carried out to evaluate PAS840’s impact on intracellular Ca^2+^ levels, oxidative stress, apoptosis rate, and MMP in glutamic acid-injured cells. q-PCR and Western blot were used to measure the expression of the *N-methyl-D-aspartate* receptor (NMDAR) pathway. The results show that PAS840 acts on multiple targets, reducing NMDAR activity, preventing Ca^2+^ influx, mitigating oxidative stress and inflammation, and enhancing cell viability. This suggests that PAS840 is a potential treatment for NMDAR-induced cell damage.

## 1. Introduction

Neurodegenerative diseases, including Alzheimer’s disease (AD), Parkinson’s disease, and Huntington’s disease, have emerged as a global health concern, exerting a profound impact on both families and society [1]. Neurodegenerative diseases have complex etiologies, involving multiple factors such as gene mutations, oxidative stress, mitochondrial dysfunction, abnormal immune responses, inflammation, and excitotoxicity. These diseases are irreversible. During disease progression, patients gradually lose their ability to perform daily activities and may ultimately die due to complications [2].

The *N-methyl-D-aspartate* receptor (NMDAR) glutamate-gated ion channel has a big impact on how people learn and remember things. It can change the structure of dendritic and axonal cells and the flexibility of synapses by changing how neurons work, which in turn, changes how neurons connect to each other [3]. During normal excitability, after the NMDAR is positively modulated by glutamic acid (Glu), the main excitatory neurotransmitter in the neuronal cells of the brain, the postsynaptic neuron undergoes a depolarization process in which Mg^2+^ ions blocking the channel are removed, allowing Ca^2+^ ions to flow through the channel and enter the nerve cell. This process has a far-reaching and significant impact on learning and memory functions. However, excessive release of glutamate can stimulate neuronal death in the mammalian central nervous system, manifesting as excitatory neurotoxicity [4]. Excessive glutamate over-activates NMDAR, promoting Ca^2+^ influx and resulting in intracellular calcium overload. This triggers mitochondrial membrane depolarization, induces changes in mitochondrial morphology and structure, and releases cytochrome C [5]. Simultaneously, it facilitates the generation of reactive oxygen species clusters (ROS), which induces downstream caspase cascade reactions [6] and ultimately leads to cell death [7]. And when excessive ROS production is induced after a noxious stimulus to the organism, the degree of oxidation exceeds the oxidant-scavenging capacity of the organism, resulting in a disruption of the balance of the antioxidant system in the body. The process is known as oxidative stress [8]. ROS-induced neuronal damage has been shown to be associated with the etiology of many common neurological disorders [9,10]. Studies have shown that oxidative stress in the brain under progressive aggravation reduces antioxidant defenses, accelerates neuronal cell senescence, and is one of the major risk factors for neurodegenerative disease [11].

*Periplaneta americana* (L.) (PA), commonly referred to as the “cockroach” [12], has been investigated in modern studies. It has been demonstrated that its extracts are rich in small molecular peptides, amino acids, nucleosides, and numerous types of organic small molecules [13,14]. Within these components, certain substances possess diverse pharmacological activities, including antioxidant, anti-inflammatory, anti-heart failure, anti-fibrotic, and tissue repair properties [15,16].

This study is based on our group’s previous finding that the PA extract (PAS840, named according to the drug extraction process in Section 2.3) can effectively protect PC12 cells from hydrogen peroxide-induced damage. In this study, taking the typical neurodegenerative disease AD as the target disease, after analyzing the material components of PAS840 by LC-MS/MS, network pharmacology was used to predict and analyze the targets of PAS840 for AD. An in vitro excitotoxic PC12 cell (cell line derived from a rat pheochromocytoma, a tumor of the adrenal medulla) model was established using Glu induction. After intervening with PAS840 in the model cells, the effects of PAS840 on cell viability, intracellular Ca^2+^ levels, oxidative stress, and NMDAR1 expression were explored, aiming to lay the foundation for subsequent studies on the impact of PAS840 on neurodegenerative diseases.

## 2. Materials and Methods

### 2.1. Instruments

A CKX41SF inverted microscope (OLYMPUS, Center Valley, PA, USA); Varisoskan LUX multifunctional enzyme labeling instrument (Thermo Fisher Scientific, Waltham, MA, USA); BD FACSCantoTM II flow cytometer (BD, Franklin Lake, NJ, USA); TCS SP8 laser confocal scanning microscope (Leica, Hesse, Germany); Optics Module PCR instrument, Mini-Protein vertical electrophoresis tank and electrophoresis apparatus (model 1658033, BIO-RAD, Hercules, CA, USA); Thermo Vanquish (Thermo Fisher Scientific, Waltham, MA, USA); and Thermo Q Exactive mass spectrometry detector (Thermo Fisher Scientific, Waltham, MA, USA) were obtained.

### 2.2. Reagents

L-Glu (Beijing Inokai Science and Technology Co., Ltd., Beijing, China, Lot No.: A21524); PC12 cell culture medium (Wuhan Pnosay, Wuhan, China, Lot No.: WH01112107-10SP); LDH, GSH-Px, GSH, MDA, ROS kits (Beijing Solepol, Beijing, China, Lot Nos., respectively, PC0020, BC0685, BC1195, BC1175, BC0025, CA1410); Total SOD Assay Kit/Calcium content color detection kit (Biyuntian Company, Lot No. S0101S, S1063S, Shanghai, China); Annexin V-FITC/PI Apoptosis Detection Kit (BD Company, Lot No. 556547, Franklin Lake, NJ, USA); PAGE Gel Rapid Preparation Kit (Yase, Lot No. PG112, Qingdao, China); Fekt Ultrasensitive ECL Luminescent Liquid (Mellen Biological Company, Lot No.: MA0186-Apr-29G, Dalian, China); RT-qPCR primers (synthesized by Shanghai Sangong Bioengineering Company, Shanghai, China); Tumor Necrosis Factor-alpha (TNF-alpha) kit, Interleukin-6 (IL-6) (Nanjing Jianjian Biological Co. Ltd., Nanjing, China, Lot No.: 20220827); rabbit anticlonal antibody GAPDH, rabbit anticlonal antibody NMDAR1, rabbit anticlonal antibody Cytc, rabbit anticlonal antibody Caspase-3, rabbit anticlonal HO-1 antibody (Cell Signaling Technology, Danvers, MA, USA, batch No. 8, 4, 8, 3, 1, respectively); and rabbit anticlonal antibody Bax, rabbit anticlonal antibody Bcl2, goat anti-rabbit IgG-HRP secondary antibody (Wuhan Sanying. Wuhan, China, Lot numbers: 00104968, 00110639, 20000483), Highly differentiated PC12 cells (Wuhan Punosai Life Science and Technology Company Limited, Wuhan, China, No. 8300L222009) were obtained.

### 2.3. Sample Preparation

*P. Americana* (L.) was purchased from Yunnan Jingxin Biotechnology Co., Ltd. (Dali, China) and identified as *P. Americana* (L.) by Prof. Yang Zizhong from the Yunnan Key Laboratory of Insect Biomedicine R&D, Dali University. The dried insect body of *P. Americana* (L.) was crushed and extracted with 5 times the mass of the insect body in 90% ethanol, and the extracted alcohol was concentrated and defatted to obtain a defatted extract, which was then chromatographed on an S-8 macroporous adsorbent resin column, washed with ultrapure water, and then eluted with 40% ethanol. The ethanol solution was then collected, concentrated under reduced pressure at 60 °C, and freeze-dried, which resulted in the sample PAS840, which was used for the analysis of LC-MS/MS and in vitro experiments.

### 2.4. Mass Spectrometry Identification of Compounds Present in PAS840

The PAS840 samples were gathered and ground using a tissue grinder (Zhejiang Meibi Technology Co., Hangzhou, China) for 120 s at 50 Hz. After grinding, the samples were vortexed for 30 s in 600 µL of methanol containing 4-chloro-L-phenylalanine (4 ppm) to facilitate extraction. The samples were then centrifuged for 10 min at 12,000 rpm following sonication at 24 ± 2 °C. The supernatant was filtered through a 0.22 μm filter membrane, and the filtrate was finally introduced into the test bottle for LC-MS/MS analysis to detect small-molecule compounds.

The ACQUITY UPLC^®^ HSS T3 (2.1 × 100 mm, 1.8 µm) (Waters, Milford, MA, USA) column was used for the analysis of the samples on a Thermo Vanquish (Thermo Fisher Scientific, USA) ultra-high-performance liquid chromatography (UHPLC) system. The column temperature was set at 40 °C, a flow rate of 0.3 mL/min, and an injection volume of 2 μL. The positive ionization mode used 0.1% formic acid in acetonitrile (B2) and 0.1% formic acid in water (A2) as the mobile phases. The gradient elution program was 0 to 1 min, 8% B2; 1 to 8 min, 8% to 98% B2; 8 to 10 min, 98% B2; 10 to 10.1 min, 98% to 8%B2; and 10.1~12 min, 8% B2. And the negative ionization mode used acetonitrile (B3) and 5 mM ammonium formate in water (A3) as the mobile phases. The gradient elution program was 0–1 min, 8% B3; 1–8 min, 8–98% B3; 8–10 min, 98% B3; 10–10.1 min, 98–8% B3; and 10.1–12 min, 8% B3. A Thermo Q Exactive Mass Spectrometry Detector (Thermo Fisher Scientific, USA) equipped with an electrospray ionization (ESI) source in the positive and negative ion modes was used to collect data separately. The auxiliary gas, sheath gas, and positive ion spray voltage were 10 arb, 40 arb, and 3.50 kV, respectively. The negative ion spray voltage was −2.50 kV. The main ion-scanning range was *m*/*z* 100–1000. The capillary temperature was 325 °C, and a resolution of 70,000 was achieved during the primary complete scan. Secondary cleavage was performed using HCD with a collision energy of 30 eV and a secondary resolution of 17,500. The first ten ions obtained were broken, and any extraneous MS/MS data were eliminated using dynamic exclusion.

In the detection of peptide components in the PAS840 samples, liquids A (100% water and 0.1% formic acid) and B (80% acetonitrile and 0.1% formic acid) were used as mobile phases. Liquid A (10 mL) was used to dissolve the lyophilized powder sample. For liquid–liquid detection, 1 µg of the supernatant was injected into the sample after it had been centrifuged for 20 min at 4 °C at 14,000 rpm The elution conditions used for liquid chromatography are listed in Table 1. Using a Nanospray FlexTM (NSI) ion source and a Q Exactive HF-X mass spectrometer, the ion spray voltage was set at 2.4 kV, the ion transfer tube temperature at 275 °C, and the mass spectrum was acquired in data-dependent mode with a full scan range of *m*/*z* 100–1500 and a resolution of 120,000 (200 *m*/*z*) for the first stage of the mass spectrum. The AGC measured 3 × 10^6^ and the maximum C-trap injection time was 80 ms. The parent ion with the highest ion intensity in the entire scan was chosen for fragmentation using the high-energy collision cleavage (HCD) method. Secondary mass spectrometry (MS) was performed at a resolution of 15,000 (200 *m*/*z*). The AGC was 5 × 10^4^, with a maximum injection time of 45 ms and the energy of peptide fragmentation and collision set at 27%. Raw data (.raw) were used for MS detection. Peptide sequences were retrieved by de novo analysis using the peptides’ de novo-only peptide databases.

### 2.5. Network Pharmacology Analysis of PAS840 Against AD

#### 2.5.1. Acquisition of Disease Targets Corresponding to PAS840 Components

The molecular details and structures for the small-molecule compounds detected using PAS840 and entered into the SwissADME database (http://www.swissadme.ch/index.php, accessed on 5 April 2024) [17] were obtained from the PubChem database (https://pubchem.ncbi.nlm.nih.gov/, accessed on 5 April 2024) [18]. Following a screening process that identified compounds with therapeutic disease target effects, columns with a probability > 0.01 were chosen, and the associated targets were observed.

To obtain the related protein information, the peptide sequences that scored highest on the database peptides analysis (−10 lg*p* value) and those that scored higher on de novo-only peptide analysis (>95 confidence level) were chosen and added to the Emboss database (https://www.ebi.ac.uk/Tools/seqstats/emboss_pepstats/, accessed on 12 April 2024). Peptide sequences with a charge value of ≥0 and an isoelectric point value of ≤12 were screened according to the conditional characteristics of food-borne anti-inflammatory peptides and neuroprotective peptides [19,20,21,22], The peptide sequence numbers were then entered into the NoverPor (https://www.novoprolabs.com/tools/convert-peptide-to-smiles-string, accessed on 12 April 2024) web tool to convert the sequence numbers to the smile format, and the similarity ensemble approach (SEA) (https://sea.bkslab.org/, accessed on 15 April 2024) was used to obtain the target protein for each peptide prediction information. After eliminating duplicates, the target protein information of the peptides and small molecules were combined to obtain the desired information.

#### 2.5.2. Alzheimer’s Disease Target Acquisition

The search terms “Alzheimer” and “AD” were selected to access the OMIM (https://omim.org/, accessed on 18 March 2024), GeneCards (www.genecards.org/, accessed on 18 March 2024), and DisGeNET (https://www.disgenet.org/search, accessed on 18 March 2024) databases to obtain the excel results of disease genes. The search results of the three databases were combined to remove the duplicate disease target information, and finally, the obtained targets of drugs and diseases were processed using Venny 2.0.1 to obtain the cross-targets of drugs and diseases.

#### 2.5.3. Disease Target Interaction Network Construction (PPI)

The protein interaction search prediction database STRING (https://string-db.org/, accessed on 3 May 2024) [23] was used to hide the nodes that were not connected to the network, and the rest of the parameters were set by default to export the relevant data. The protein–protein interaction (PPI) network was analyzed using the Centiscape 2.2 plugin in Cytoscape 3.9.1 software, based on the default filters of Closeness (>5.158153796225441 × 10^−4^), Betweenness (>1475.2112149532836), and Degree (>8.370093457943925). The core protein target information was obtained, and then, Network Analyzer was used to calculate the node Degree to evaluate the roles of different protein targets and differentiate them with different colors or sizes, respectively.

#### 2.5.4. GO and KEGG Enrichment Analysis

The core protein target information was imported into the Metascape (https://metascape.org/, accessed on 3 May 2024) database and the DAVID (https://david.ncifcrf.gov/, accessed on 3 May 2024) database, respectively, and OFFICIAL_GENE_SYMBOL was selected in DAVID to perform gene ontology (GO) biological process analysis and Kyoto encyclopedia of genes and genomes (KEGG) pathway enrichment analysis, respectively. The enrichment results were then visualized using the online bioinformatics platform Microbiosense (http://www.bioinformatics.com.cn/, accessed on 3 May 2024).

#### 2.5.5. Active Ingredient–Pharmacological Target Network Construction

With the obtained information on the constituent targets, disease intersection targets, and KEGG pathway targets, the network worksheet of constituent-pharmacodynamic target–disease pathway was created. The information was integrated into a Tape file, and the data were finally imported into the Cytoscape 3.9.1 software to construct the constituent-pharmacodynamic target–disease pathway network diagram. At the same time, Centiscape 2.2, a plug-in for Cytoscape 3.9.1, was utilized to calculate the node Degree values to evaluate the relationships among the sample components, pharmacodynamic targets, and disease pathways.

#### 2.5.6. Molecular Docking

To verify the binding activity between core target proteins and active ingredients, molecular docking was performed using the software AutoDock Vina 1.5.7 [24]. The molecular docking was performed for the target proteins with the highest degree of association screened in the PPI network and the corresponding active ingredients. The structure files of the target proteins and active ingredients were obtained through the PDB (https://www.rcsb.org/, accessed on 8 May 2024) database. The search space volume was >27,000 Angstrom^3^ (see FAQ) (use random seed files to refer to the raw data), and the affinity value was recorded. Then, the proteins were de-watered and de-liganded using PyMoL 4.6.0 software, and the active sites were selected for molecular docking centered on the original ligands of the target proteins. And the PyMoL software was selected to obtain the visualization results.

### 2.6. Effect of PAS840 on the Viability of PC12 Cells

PC12 cells in the logarithmic growth phase were harvested and subsequently diluted to form a cell suspension with a density of 1 × 10^5^ cells/mL. The resulting cell suspension was then inoculated into 96-well plates, with 100 μL of the suspension per well. The plates were cultured at 37 °C in an environment containing 5% CO_2_ for 24 h. After the initial incubation, 50 μL of PAS840 medium at final concentrations of 8, 16, 32, 64, 128, 256, and 512 μg/mL were added to the respective wells. In the naive group, specifically, only 50 μL of the medium were introduced. The cells were then incubated for an additional 24 h. Subsequently, the cell morphology was observed under an inverted microscope, and pictures were taken for documentation. Next, 10 μL of Cell Counting Kit-8 (CCK-8) were added to each well. After a reaction period of 1 h, the absorbance (OD) value of each group at 450 nm was measured using a multifunctional microplate reader. Based on these measurements, the survival rate of the cells (%) was calculated, thereby determining the safe dosage of PAS840.

### 2.7. Establishment of PC12 Cell Injury Model

Cultured and inoculated cells were placed into 96-well plates, as described in Section 2.6. The cell morphology was observed under an inverted microscope and image. Subsequently, in the model group, Glu solutions at concentrations of 20, 30, 40, and 50 mmol/L were added, and in the naive group, an equal volume of PBS was added. After a further 24 h incubation, the cell morphology was observed under the inverted microscope, and pictures were taken for recording. Then, 10 μL of CCK-8 were added to each well. After a 1 h reaction, the OD value of each group was measured at 450 nm using a multifunctional microplate reader. The cell viability (%) was calculated to determine the appropriate Glu concentration for modeling purposes.

### 2.8. Effect of Experimental Grouping and PAS840 on the Survival Rate of Damaged Cells

The cells were counted at the logarithmic growth phase in a 96-well plate. The cell density was adjusted to 1 × 10^5^ cells/mL, 100 μL/well. In the 6-well plate, the cell density was adjusted to 3.5 × 10^5^ cells/mL, 2 mL/well, and cultured for 24 h at 37 °C with 5% CO_2_, then grouped into different treatments: PAS840-L, PAS840-M, and PAS840-H. After the culture was completed, the cell morphology was observed under an inverted microscope and imaged for recording, and the OD values of each group at 450 nm were detected by a multifunctional microplate reader after adding the CCK-8 reaction for 1 h. The survival rate (%) of the cells was calculated.

### 2.9. Effects of PAS840 on SOD, GSH-Px, GSH, and MDA Levels in Damaged Cells

After the cells were cultured and grouped as described in item 2.8, the media solution was removed, and the cells from each group were collected. Subsequently, 100 μL of cell lysate were added to ensure complete cell lysis. The mixture was then centrifuged at 1200 rpm/min for 5 min, and the supernatant was collected. The samples were then detected following the kit instructions, with the OD values of the samples from each group being measured by a multifunctional microplate reader at the specified wavelength. Record the detected OD values, and analysis should be performed in accordance with the kit instructions (the test instruction manual of the kit has been uploaded to the Appendix A).

### 2.10. Effect of PAS840 on ROS in Damaged Cells

The DCFH-DA probe was employed to facilitate the entry of the probe into the cell membrane and interact with intracellular ROS, thereby generating strongly fluorescent DCF. The fluorescence intensity is positively correlated with the production of ROS. Cells were cultured and grouped according to “2.8” (ROS-positive control—Rosup group was set at the same time). Then, the culture medium was removed, 1.5 mL DCFH-DA with a final concentration of 10 μmol/L was added to each well, and incubated in an incubator at 30 °C for 20 min under dark light. The well plates were removed and mixed every 5 min. Make the probe fully contact the cell. After incubation, the DCFH-DA was discarded, and the cells in each group were washed twice with serum-free 1640. Then, the cell suspension was collected and prepared. One hundred μL of cell suspension were taken from each group into the 96-well plates. The fluorescence intensity (ROS-RFU) of the ROS in each group at an excitation wavelength of 488 nm and an emission wavelength of 525 nm was detected by a multifunctional microplate reader (the test instruction manual of the kit has been uploaded to the Appendix A).

### 2.11. Effect of PAS840 on Ca^2+^ in Damaged Cells

After the cells were processed according to the culture grouping under “2.8”, remove the medium, collect the cells of each group, and add 150 μL of cell lysate, respectively. Once fully lysed, centrifuge at 1200 rpm/min for 5 min, take 50 μL of supernatant in a 96-well plate, refer to the operation instructions of the Ca^2+^ test kit, and add 150 μL of working liquid in the kit to the 96-well plate (the test instruction manual of the kit has been uploaded to the Appendix A). Incubate for 8 min at room temperature and avoid light. Then use a multi-functional enzyme marker to detect the OD value at 575 nm and calculate the Ca^2+^ concentration of each group according to the Ca^2+^ standard curve.

### 2.12. Effect of PAS840 on the Release of LDH from Damaged Cell Cultures

After the cells were processed according to the culture group under “2.8”, collect the cell culture medium of each group, centrifuge at 1200 rpm/min for 5 min, take 10 μL of supernatant from each group, and add it into a 96-well plate. Follow the instructions of the LDH activity quantitative assay kit (the test instruction manual of the kit has been uploaded to the Appendix A). Then, leave it at room temperature for 3 min and, then, detect the OD value of each group at 450 nm with a multifunctional enzyme marker. After 3 min at room temperature, the OD value of each group was measured by a multifunctional enzyme-labeling instrument at 450 nm. The data were recorded, and the amount of LDH released was calculated.

### 2.13. Effect of PAS840 on the Apoptosis Rate of Damaged Cells

After the cells were treated according to the culture group under “2.8”, the cell culture medium and the cells of each group were collected in the same centrifuge tube, centrifuged at room temperature for 5 min at 1200 r/min, and the supernatant was discarded. The cells were gently resuspended in PBS and washed twice, then centrifuged, and the supernatant was discarded. A 1× binding solution (the fully prepared working solution in the kit; the test instruction manual of the kit has been uploaded to the Appendix A) was added to gently resuspend the cells, and then, the cells were incubated with Annexin V-FITC and PI staining solution at room temperature and protected from light for 15 min. The single-positive and double-positive tubes were also set up for the Annexin V-FITC and PI staining solution. The cells were detected on a flow cytometer immediately after the incubation was completed.

### 2.14. Effect of PAS840 on MMP in Damaged Cells

After the cells were processed according to the culture group under “2.8” (a positive control group with decreased mitochondrial membrane potential was also set up: CCCP group), the culture medium was removed, and 1.5 mL of JC-1 staining working solution was added to each well (5 μL of JC-1 and 0.8 mL of ultrapure water were vigorously vortexed and mixed thoroughly, and 0.2 mL of 5 μL of 5 × JC-1 staining buffer was added and mixed thoroughly) and incubated for 20 min in an incubator protected from light. After incubation, discard the working solution and, then, wash the adherent cells with JC-1 staining buffer (1×) 2 times. Then, add 1 mL of cell culture medium. Three fields of view were randomly selected for each group, and they were then observed and imaged with a laser confocal scanning microscope (green fluorescence to detect JC-1 monomer; red fluorescence to detect JC-1 polymer). At the same time, collect the cells of each group to prepare the cell suspension and use multi-functional enzyme markers to detect the fluorescence value A1 of the JC-1 monomer (excitation wavelength 488 nm, emission wavelength 530 nm) and the fluorescence value A2 of the JC-1 polymer (excitation wavelength 529 nm, emission wavelength 590 nm). Calculate the fluorescence value A2 of the polymer and the fluorescence value A2 of the polymer. The fluorescence value A2 of the JC-1 monomer (excitation wavelength 488 nm, emission wavelength 530 nm) and the fluorescence value A2 of the JC-1 polymer (excitation wavelength 529 nm, emission wavelength 590 nm) were calculated, and the fluorescence ratio A (A = A2/A1) of each group was calculated to quantify MMP.

### 2.15. Effect of PAS840 on mRNA Levels of Damaged Cell-Related Proteins

After the cells were cultured and grouped according to item 2.8, the culture medium was removed, and the cells were collected. One mL of RNAiso Plus was added to each group of cells to extract the total RNA. The purity and concentration of RNA were detected by an ultraviolet spectrophotometer, and the cDNA was synthesized by reverse transcription and amplified by RT-qPCR after the concentration of sample RNA was unified. GAPDH was used as the internal reference, and the target gene was quantitatively analyzed by 2^−ΔΔCt^. The primers are shown in Table 2.

### 2.16. Determination of TNF-α and IL-6 in Culture Medium of Each Group by ELISA

After the cells were cultured according to “2.8”, the cell culture medium of each group was collected, and the supernatant was centrifuged as the sample. The kit was equilibrated at room temperature for 30 min; 50 μL of supernatant were taken from each group. The OD values of each group were measured at 450 nm using a multifunctional enzyme labeling instrument, and the data were recorded. The results were calculated by ELISAcala, and the fitting model was chosen logistically (four parameters) to draw the standard curve. The levels of TNF-α and IL-6 factors in each group were calculated according to the standard curve.

### 2.17. Detection of Related Proteins by Western Blot Assay

After the cells were grouped according to “2.8”, the culture medium was removed, and 200 μL of lysate were added to the cells of each group. The supernatant was collected by centrifugation after lysis at 4 °C for 30 min. The protein concentration was detected by the BCA (bicinchoninic acid assay) method, and 20 μg of the sample were quantified. The protein samples were separated by 10% SDS-PAGE gel electrophoresis, and the separated protein samples were transferred onto a PVDF membrane, which was closed with a 5% BSA sealing solution for 2.5 h. After the sealing was completed, the membrane was washed three times, and NMDAR1 (1:1000), Cytc (1:1000), Bax (1:5000), and Bcl2 (1:1000) were added, along with Capsase-3 (1:1000), and incubated at 4 °C overnight. The primary antibody was discarded. The secondary antibody was added and incubated for 1 h at room temperature on a shaking table. The color was developed, and the bands were observed. The grayscale value of each band was calculated by ImageJ 1.8.0 software and compared with that of GAPDH for semi-quantitative analysis.

### 2.18. Statistical Analysis

SPSS 22.0 statistical software was used to analyze the data; the measured data were expressed as x ± s. Comparisons between multiple groups were made by one-way ANOVA, and two-by-two comparisons were made by a Tukey *t*-test; *p* < 0.05 was used to indicate that the difference was statistically significant.

## 3. Results

### 3.1. LC-MS/MS Analysis and Active Ingredient Screening

By an LC-MS/MS analysis, the PAS840 contained 223 small-molecule compounds and 16,788 peptide sequences, of which the database peptides had a total of 2035 peptide sequences and the de novo-only peptides had a total of 14,753 peptide sequences. According to the set peptide-screening conditions, 50 peptide sequences were obtained from the top 106 peptide sequences having a −10 lg*p* value in database peptides, and 128 peptide sequences were obtained from 259 peptide sequences with a confidence level greater than 95 in de novo-only peptides, totaling 178 peptide sequences.

### 3.2. Disease Gene Acquisition and Screening

A total of 999 AD-related targets were obtained from the OMIM, GeneCards, and DisGeNET databases, and 25,522 disease targets were finally obtained after merging duplicate targets. After plotting the Venn diagram of the PAS840 active ingredient targets and the AD targets and analyzing them, 720 intersecting targets were finally obtained (Figure 1).

### 3.3. PPI Network Construction and Core Protein Screening

The 720 pharmacodynamic targets were imported into the STRING database for protein interaction analysis, removing the non-intersecting REG1A, and 578 targets with 2275 linkages remained, from which 86 key core protein targets with 519 linkages were screened out of the target network map, among which were SRC, STAT3, MAPK1, BRCA1, and JUN (core proteins closely related to inflammatory response), and HSP90AA1, ESR1, GRB2, Akt1, EGFR, and FYN (core proteins with positive effects on AD treatment and cell growth promotion), all of which have strong correlations with AD (Figure 2).

### 3.4. Results of Rnrichment Analysis

The GO analysis included biological process (BP), cell composition (CC), and molecular function (MF), and a total of 1563 results were obtained, of which 1067 were for biological function, 168 were for cell composition, and 328 were for molecular function. The top 10 were visualized according to the *p*-value (Figure 3A). A GO analysis showed that the targets of core protein action in the biological process were mainly focused on inflammatory response and anti-stimulation response, positive regulation of cytosolic calcium ions, and positive regulation and anti-apoptotic effects. The process of cellular composition mainly involves the plasma membrane, presynaptic membrane, postsynaptic membrane, dendrites, cell surface, and membrane vesicles, indicating that the disease-related active components of PAS840 mainly act in the membrane-like components of the cell and have an important role in the regulation of neurotransmission and transmembrane transport. In terms of molecular function, it mainly acts on protein binding, signaling, and G protein-coupled receptor signaling processes.

The KEGG enrichment analysis was exported to the analysis results, and a total of 178 pieces of KEGG pathway information was obtained. The top 20 entries related to the signaling pathways were selected according to the size of the *p*-value for the bubble map visualization analysis (Figure 3B). According to the judgment that the larger the bubble, the darker the color and the higher the correlation, it showed that the pathways with the highest correlation were concentrated in the cancer pathway and the neuroreceptor–ligand pathway. The action of the calcium regulatory pathway and the renin–angiotensin pathway to regulate the blood–brain barrier can produce a positive regulatory effect in anti-apoptosis, anti-inflammatory, and endocrine regulation, and other pathways have a close connection.

### 3.5. Ingredient–Target–Pathway Interaction Network Analysis

First, we sorted all compounds in PAS840 by the first letter of their names. The ID names were named PA + sorting number based on the sorting results, and network files were made according to the pharmacodynamic disease target and KEGG ID of each active ingredient. The tape files were summarized and imported into Cytoscape 3.9.1 software to construct the active ingredient–target–pathway interaction network diagram (Figure 4), respectively, and the nodes with no linkage significance were deleted, which showed that there was a very high degree of association between PAS840 and AD treatment targets. The linkage was high.

### 3.6. Results of Molecular Docking

According to the binding energy after molecular docking, the lower the binding energy of the ligand and receptor, the more stable the structure and the stronger the binding activity. The top 10 disease targets with good correlation in the PPI analysis were selected, and the 10 small molecules with the highest correlation with the disease targets were selected to be molecularly docked with 10 peptide sequences (Table 3 and Table 4). The results of the docking showed that 97% of the small molecules had a binding energy of <−5 kcal/mol to their targets, and 35% of the components had a binding energy of <−7.0 kcal/mol to their targets. In the peptide sequence, 100% of the components had binding energies < −5 kcal/mol, and 51% of the components had binding energies < −7.0 kcal/mol to their targets. The docking results were visualized in a heat map (Figure 5A), which indicated that all of the components of PAS840 have strong binding activities to their targets, especially the small molecules, such as PA3, PA4, PA35, and the peptide sequences, such as peptide51, peptide108, and peptide114, which have higher binding energies to the core targets. The binding of SRC, EGFR, and MAPK1 to the active components of PAS840 showed high activity. The docking results of small molecules to the core components of proteins in the peptide sequences with minimal binding energies were visualized and shown with PyMoL software (Figure 5B).

### 3.7. Effect of PAS840 on the Survival Rate of PC12 Cells

By setting different concentrations of PAS840 on PC12 cells after 24 h, compared with the naive group (negative control group without any treatment), when the concentration of PAS840 was 8, 16, 32, 64, or 128 μg/mL, there was no significant effect on the cell survival rate. But, when the concentration of PAS840 was 512 μg/mL, the survival rate of the cells was significantly reduced (*p* < 0.0001) (Figure 6).

### 3.8. Modeling of Excitatory Injury in PC12 Cells

When the cells were stimulated by Glu, the morphology appeared crumpled, and the synapses disappeared (Figure 7). Compared with the naive group, the cell survival rate gradually decreased with the increase in the Glu concentration (Figure 7). When the Glu concentration was 30 mmol/L, it was convenient to observe the cell state at this concentration, and the cell survival rate was (50.20 ± 1.69)%. The cell damage state at this concentration better reflected the protective effect of the samples on the damaged cells, so we determined that the Glu concentration of 30 mmol/L was the subsequent model stimulation concentration. Therefore, 30 mmol/L Glu was determined as the subsequent stimulation concentration for the model.

### 3.9. Protective Effect of PAS840 on Injured Cells

After 24 h induction by Glu, compared with the Con group (negative control group without any treatment), the cell number and survival rate in the Mod group (Use 30 mmol/L Glu as the modeling concentration for the model group and each administration group) were significantly decreased (*p* < 0.01), the levels of SOD, GSH-PX, and GSH were significantly decreased (*p* < 0.001), and the level of MDA was significantly increased (*p* < 0.01). After PAS840 treatment, compared with Mod, the cell morphology was significantly restored to a nearly normal cell state with the increase in the dose (Figure 8), and the cell survival rate was significantly increased (*p* < 0.0001) in a dose-dependent manner (Figure 8). And the SOD (*p* < 0.001), GSH-PX (*p* < 0.001), and GSH (*p* < 0.01) levels were significantly increased at the same time. The level of MDA was significantly decreased (*p* < 0.01) (Figure 8).

### 3.10. Effect of PAS840 on ROS and Ca^2+^ Content of Damaged Cells

The fluorescence intensity of the ROS and Ca^2+^ content in the cells was significantly increased after Glu induction compared with the Con group (*p* < 0.05). After the PAS840 intervention, the fluorescence intensity of the ROS and Ca^2+^ content was reduced in a dose-dependent manner. The fluorescence intensity of the ROS in the medium-dose group and the high-dose group decreased significantly (*p* < 0.001). The content of Ca^2+^ ions decreased significantly in the high-dose group (*p* < 0.05) (Figure 9).

### 3.11. Effect of PAS840 on Apoptosis and MMP in Injured Cells

Compared with the Con group, the amount of LDH leakage in the cells induced by Glu was significantly increased (*p* < 0.0001), and the confocal and flow cytometry results showed that the apoptotic cells in the Mod group were significantly increased (Figure 10 and Figure 11). After the intervention of PAS840, both LDH leakage and the apoptosis rate were reduced in a dose-dependent manner, and MMP was significantly increased in the high-dose group (*p* < 0.0001) (Figure 11).

### 3.12. Effect of PAS840 on the Expression of NMDAR1, Cytc, Bcl-2, Bax, Caspase-3, TNF-α, and IL-6 mRNA in Damaged Cells

According to the RT-qPCR results, the mRNA expression of NMDAR1 (*p* < 0.0001), Cytc (*p* < 0.001), Bax (*p* < 0.01), Caspase-3 (*p* < 0.001), TNF-α (*p* < 0.0001), and IL-6 (*p* < 0.0001) in the cells induced by Glu was significantly increased, and the mRNA expression of Bcl-2 was significantly decreased (*p* < 0.05). The cells were treated with PAS840 and, when compared with the Mod group, it could significantly decrease the mRNA expression of NMDAR1 (*p* < 0.01), Cytc (*p* < 0.001), Bax (*p* < 0.0001), Caspase-3 (*p* < 0.0001), TNF-α (*p* < 0.0001), and IL-6 (*p* < 0.0001) and significantly elevate the mRNA expression of Bcl-2 (*p* < 0.001) (Figure 12).

### 3.13. Effect of PAS840 on the Expression Levels of TNF-α and IL-1β Factors in the Culture Fluid of Damaged Cells

According to the results of the ELISA, Glu can significantly increase the expression level of TNF-α (*p* < 0.001) and IL-1β (*p* < 0.0001) factors in the PC12 cell culture fluid. After treatment with PAS840, it can significantly reduce the expression level of the above inflammatory factors compared with the Mod group (Figure 13).

### 3.14. Effect of PAS840 on Protein Expression of NMDAR1, Cytc, Bcl-2, Bax, and Caspase-3 in Injured Cells

PC12 cells induced by Glu significantly increased the protein expression of NMDAR1 (*p* < 0.05), Cytc (*p* < 0.05), Bax (*p* < 0.01), and Caspase-3 (*p* < 0.05) and significantly decreased the protein expression of Bcl-2 (*p* < 0.001) compared with the Con group. After treatment with PAS840, compared with the Mod group, it could significantly decrease the NMDAR1 (*p* < 0.05), Cytc (*p* < 0.01), Bax (*p* < 0.01), and Caspase-3 (*p* < 0.05) protein expression and significantly elevate Bcl-2 protein expression (*p* < 0.001) (Figure 14, WB images are included in the Appendix A).

## 4. Discussion

Excitotoxicity was initially proposed in 1969. It is a pathological process that is triggered by the excessive or prolonged activation of excitatory amino acid receptors and has the potential to lead to various neurological disorders, such as Alzheimer’s disease [25,26]. Thus, controlling excitotoxicity is crucial for controlling neurodegeneration and protecting neural functions. Glutamate (Glu) is an important excitatory amino acid in the central nervous system. It is released from vesicles of glutamatergic neurons upon stimulation and functions by binding to *N-methyl-D-aspartate* receptors (NMDAR). Glu exists in about one-third of central nervous system (CNS) synapses, maintaining synaptic stability and being vital for synaptic plasticity. It influences neuronal differentiation, migration, growth, and survival [27]. Therefore, Glu plays an important role in the early neurotransmission of neurological diseases like Alzheimer’s disease.

The PC12 cell line originates from the monoclonal transplantation of rat adrenal pheochromocytoma to the adrenal gland and shows neuronal-like functional characteristics after differentiation, being widely used in in vitro neurobiology and neurological disease research. In this study, we employed liquid chromatography–tandem mass spectrometry (LC-MS/MS) technology to analyze the material basis of the Periplaneta americana extract PAS840. Subsequently, taking the typical neurodegenerative disease AD as the target disease, we used network pharmacology to perform AD target docking for the components of PAS840, and the results indicated that PAS840 contains various active ingredients with potential anti-AD properties. For this reason, we established an in vitro excitotoxic cell model by inducing PC12 cells with glutamate. After the PAS840 intervention, we explored PAS840’s protective effect and potential mechanism against the excitotoxicity caused by glutamate from aspects such as cell survival rate, intracellular Ca^2+^ level, oxidative stress level, and the influence on NMDAR1 expression, hoping to provide a basis for PAS840’s research on combating neurological diseases at the animal level.

Network pharmacology, a tool capable of analyzing the effect of complex mixtures on diseases in multiple dimensions, perspectives, and levels, has been widely used in the study of traditional Chinese medicine pharmacodynamics. The PA contains abundant components such as amino acids, nucleosides, polysaccharides, peptides, and fatty acids [13,14]. Previous studies have shown that the active ingredients in PA have excellent anti-inflammatory, antioxidant, tissue repair-promoting, vasodilatory, anti-hypertensive, and angiogenesis-promoting effects [15,16]. Based on these favorable pharmacological actions of PA, in this study, LC/MS was used to detect the components of PAS840, and with AD as the target disease, network pharmacology was employed to predict its effect on AD. The results showed that PAS840 contains components that are involved in typical inflammatory target proteins, such as MPKA1/IL-6/IL-1β/CASP3 [28,29], as well as target proteins that regulate cell proliferation and apoptosis, like EGFR/AKT1/STST3.

Post-inflammatory sequelae of the brain are key pathogenic factors in AD development [30], and inhibiting brain inflammation is crucial for protecting neurological function, maintaining the blood–brain barrier, and preventing AD deterioration due to brain inflammation [31,32]. In addition, modern research has found that the SRC family tyrosine kinase is a core target receptor for neurodegenerative diseases [33], and the genetic polymorphism of the estrogen receptor (ESR) also has a strong correlation with AD [34], making it a potential therapeutic intervention target for neurodegenerative diseases. By inhibiting the activation of tyrosine kinase and regulating the BARHL1–ESR1 axis to reduce β-amyloid processing, the progression of neurodegenerative diseases, such as AD, can be slowed down [35].

Pathway enrichment analysis shows that PAS840 is mainly involved in processes such as calcium channel regulation, inflammatory response, apoptosis control, and G-protein coupling. It actively regulates the cell’s resistance to stimulatory damage, inflammatory response, neural signal regulation, and anti-apoptosis. It also affects various disease pathways, including immune disorders and toxoplasmosis. Among them, neural receptor–ligand interactions, calcium channels, and the renin–angiotensin system are closely related to neurological diseases. Calcium channel regulation is crucial for neural signal transduction; abnormal calcium accumulation in the pineal gland can cause ectopic calcification, leading to AD [36]. Renin–angiotensin may also unveil novel molecular targets for rectifying memory pathways, cerebrovascular function, and other AD-related issues [1]. The renin–angiotensin system, which controls blood pressure, can slow down the onset and progression of AD. These results are highly consistent with those of core protein analysis.

Inflammatory response is of significant importance in pathways related to neurodegenerative diseases. Up to now, two major classes of Glu receptors have been identified: ionotropic receptors, including NMDAR, kainic acid receptor (KAR), and α-amino-3-hydroxy-5-methyl-4-isoxazole receptor (AMPAR). They can couple with ion channels to form receptor-associated channel complexes that are essential for signal transduction [37]. NMDAR is the predominant subtype of Glu receptors and is crucial for initial neuronal signal conduction [38]. It is a heterotetramer usually composed of two GluN1 (NMDAR1) and two GluN2 (NMDAR2) subunits, among which NMDAR1 plays a major role [39,40,41]. NMDAR1 can induce long-term synaptic potentiation, which is a key phenomenon of enhanced synaptic transmission efficacy and a manifestation of synaptic plasticity closely associated with brain learning and memory formation [42,43].

Research indicates that NMDAR1 couples with calcium ion channels to form receptor–ion channel complexes, which mediate intercellular signal transduction. Glutamate-induced overactivation of NMDAR1 can lead to a massive Ca^2+^ influx, resulting in calcium overload and triggering downstream pathways, thereby causing neuronal damage [44]. The use of NMDAR antagonists can be used for the treatment of moderate-to-severe AD caused by synaptic damage involving NMDAR [45]. In our study, PAS840 significantly mitigated the activation of NMDAR1, reduced the influx of Ca^2+^, and decreased the cell apoptosis rate, which is similar to Koh’s finding that blocking NMDAR and reducing Ca^2+^ influx can decrease neuronal death during excitotoxicity [46]. Here, the representative drug pethidine is a low-affinity NMDAR antagonist that can reduce Glu excitotoxicity and has a neuroprotective effect on traumatic brain injury [47]. RT-qPCR and ELISA assays showed that the levels of inflammatory factors TNF-α and IL-6 in Glu-induced PC12 cells significantly increased, but these levels were markedly reduced after PAS840 intervention, indicating that Glu-induced neuroinflammation was alleviated.

Mitochondria, in addition to being the primary energy source of neurons, serve as an important buffer for cytoplasmic Ca^2+^, inhibiting excessive and prolonged elevation of Ca^2+^ [48]. Inhibition of mitochondrial Ca^2+^ uptake or disruption of mitochondrial structure and function can severely impair the clearance of cytoplasmic Ca^2+^, leading to its accumulation and subsequent cell damage, such as oxidative damage and apoptosis [49]. Studies have shown that mitochondrial Ca^2+^ uptake buffers the massive Ca^2+^ influx caused by intense Glu receptor stimulation [50,51]. Continuous Glu receptor activation increases Ca^2+^ influx, reduces MMP, impairs ATP production, increases ROS production, and disrupts the function of other organelles, all of which are closely related to neuronal death. In our research results, after PC12 cells were induced by Glu, intracellular Ca^2+^, ROS, and malondialdehyde (MDA) levels significantly increased, while MMP and antioxidant indicators, such as SOD, GSH-Px, and GSH, significantly decreased, indicating the successful establishment of a Glu-induced PC12 cell injury model. However, after treatment with PAS840, intracellular Ca^2+^, ROS, and MDA levels significantly decreased, antioxidant indicator levels increased, Bcl-2 expression was upregulated, Bax expression was inhibited, mitochondrial morphology and function were protected, the release of Cytc due to mitochondrial membrane rupture was reduced, and further, Caspase-3 expression was inhibited. This indicates that PAS840 intervention can effectively prevent Glu-induced mitochondrial damage, reduce ROS-mediated oxidative damage, enhance cellular antioxidant capacity, and inhibit apoptosis.

In conclusion, PAS840 can protect nerve cells by inhibiting the activity of the NMDAR1 receptor, counteracting excessive Ca^2+^ influx caused by the overactivation of NMDAR1, protecting mitochondrial morphology and function, alleviating oxidative stress damage, suppressing neurotoxicity, and preventing neuronal cell damage and death. These results suggest that PAS840 may have potential preventive or therapeutic effects on neurological diseases, which will provide scientific support for our next step of exploring the impact of PAS840 on AD and other neurological diseases at the in vivo experimental level.

## 5. Conclusions

Network pharmacology and in vitro cell experiments demonstrated that PAS840, an extract of *Periplaneta americana* (L.), can effectively inhibit cell damage induced by NMDAR cytotoxicity. It can effectively regulate multiple aspects and targets related to AD disease and potently inhibit inflammation and oxidative stress damage in PC12 cells, attenuate cellular damage, improve cell survival, reduce Ca^2+^ inward flow, and regulate cellular NMDAR pathway expression. This experiment demonstrated the good potential of PAS840 for neurological diseases, and it is a drug with great potential that deserves to be further investigated.

## Figures and Tables

**Figure 1 biology-14-00193-f001:**
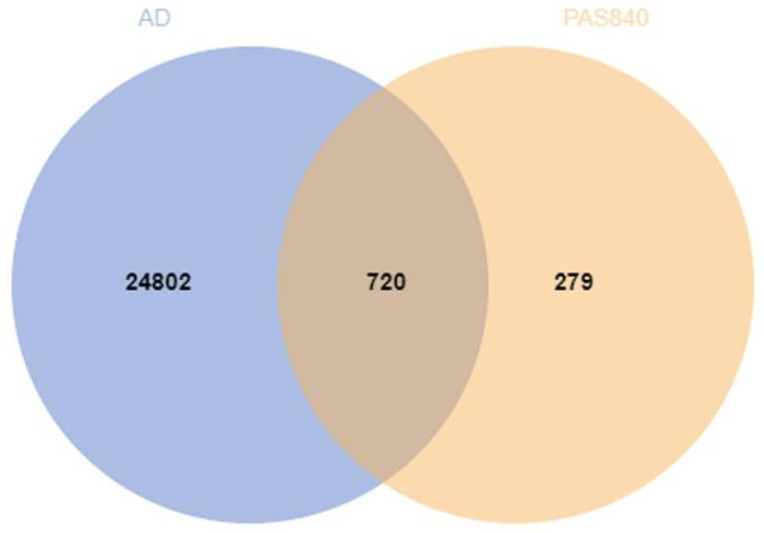
Active ingredient target–disease target Venn diagram.

**Figure 2 biology-14-00193-f002:**
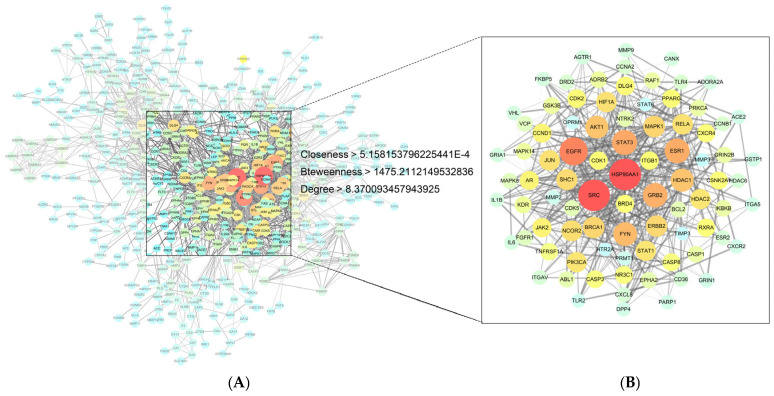
Pharmacological target analysis of PAS840 component networks. (**A**) Intersecting protein PPI network diagram; (**B**) Core protein PPI network diagram. Darker node colors indicate higher degree values, and darker and denser lines indicate higher protein associations.

**Figure 3 biology-14-00193-f003:**
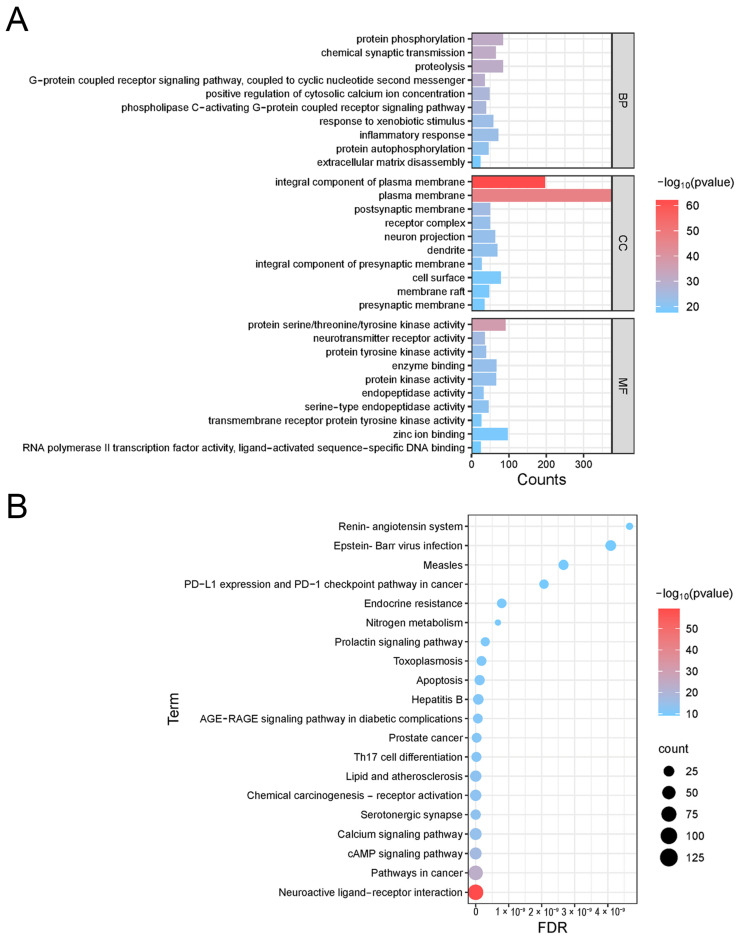
Enrichment analysis. (**A**) Column diagram of GO enrichment analysis; (**B**) Bubble map of KEGG enrichment analysis.

**Figure 4 biology-14-00193-f004:**
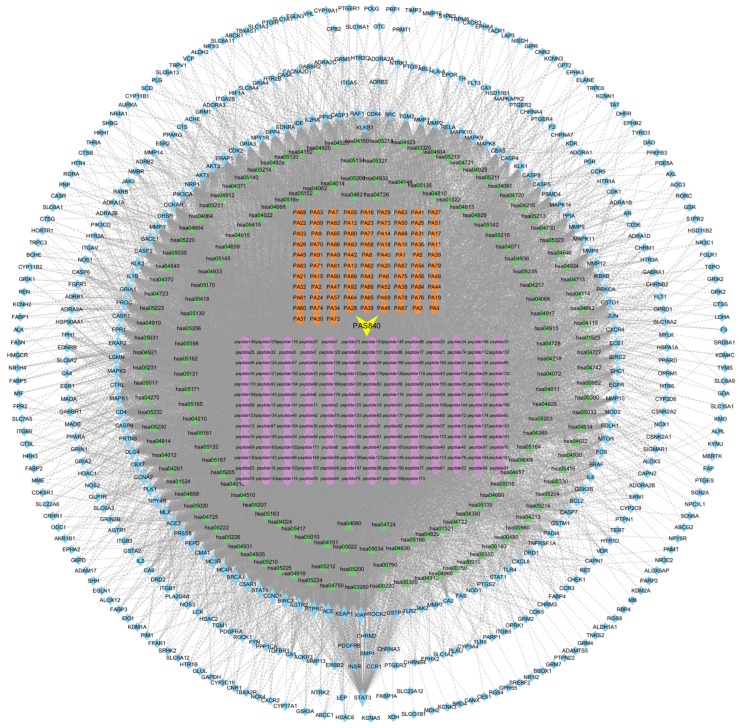
Component–target–pathway interaction network diagram. Disease targets in blue, small molecule compounds in orange, peptides in purple, and KEGG pathway in green.

**Figure 5 biology-14-00193-f005:**
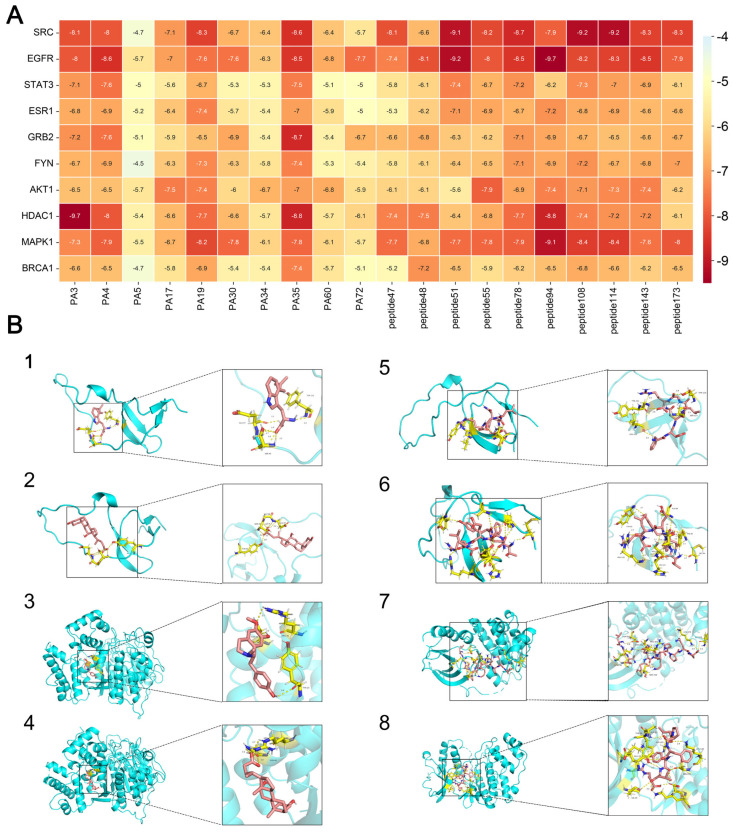
Heatmap of docking results between disease targets and active ingredients. (**A**) Heatmap of molecular-docking scores. (**B**) B1 indicates SRC docked with PA19. B2 indicates SRC docked with PA35. B3 indicates HDAC1 docked with PA3. B4 indicates HDAC1 docked with PA35. B5 indicates SRC docked with peptide108. B6 indicates SRC docked with peptide114. B7 indicates EGFR docked with peptide51 docking. B8 indicates MAPK1 docking with peptide94. In B, the blue part is the main structure of the protein, the flesh - colored part represents the structure of the drug involved in molecular docking, and the yellow part is the binding site between the drug and the protein.

**Figure 6 biology-14-00193-f006:**
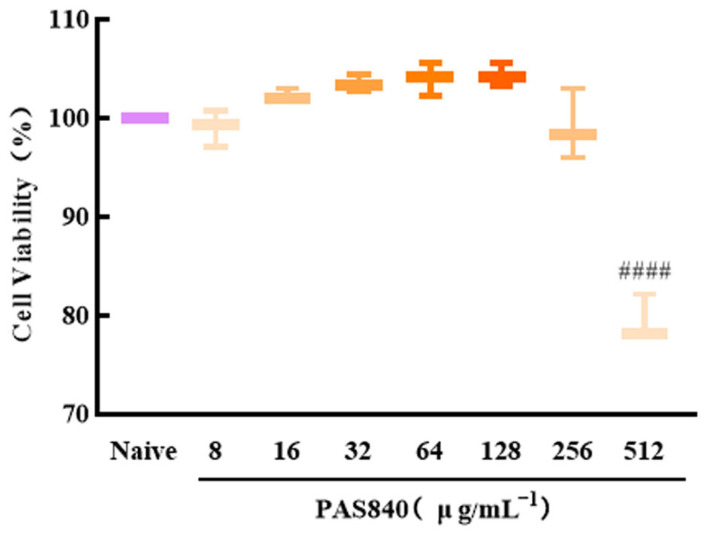
Effects of PAS840 on viability of PC12 cells. Mean ± median. *n* = 6. ^####^
*p* < 0.0001 vs. Blank group.

**Figure 7 biology-14-00193-f007:**
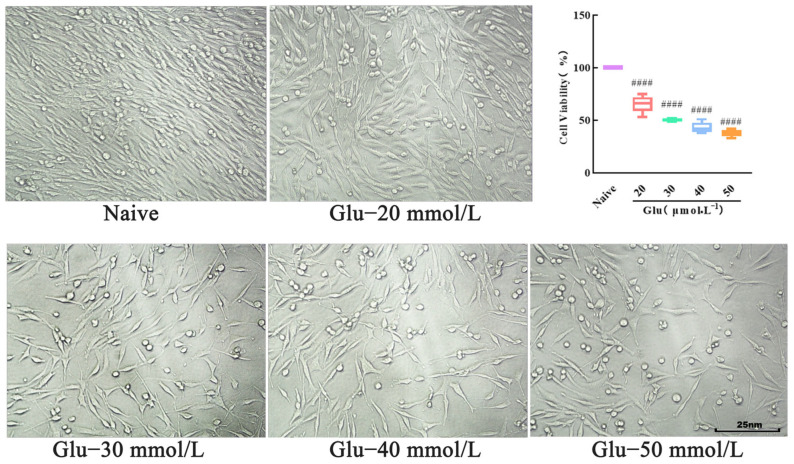
Effects of different concentrations of Glu on viability of PC12 cells. Mean ± median. n = 6. ^####^
*p* < 0.0001 vs. naive group.

**Figure 8 biology-14-00193-f008:**
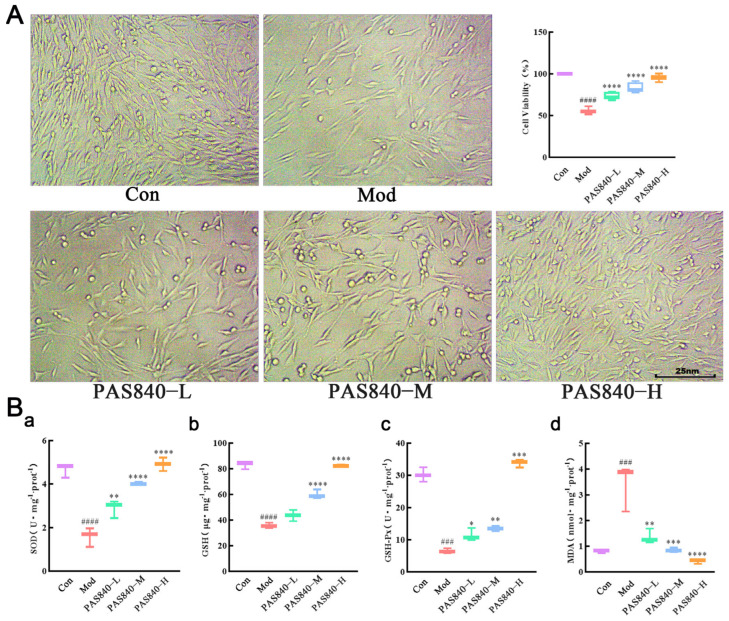
Protective effect of PAS840 on injured cells. (**A**) Effects of different concentrations of PAS840 on the viability of injured cells. (**B**(**a**)) Effect of PAS840 on SOD in injured cells. (**B**(**b**)) Effect of PAS840 on GSH-Px in injured cells. (**B**(**c**)) Effect of PAS840 on GSH in injured cells. (**B**(**d**)) Effect of PAS840 on MDA in injured cells. Mean ± median. n = 3. ^###^
*p* < 0.001, ^####^
*p* < 0.0001 vs. Con group; * *p* < 0.05, ** *p* < 0.01, *** *p* < 0.001, **** *p* < 0.0001 vs. Mod group.

**Figure 9 biology-14-00193-f009:**
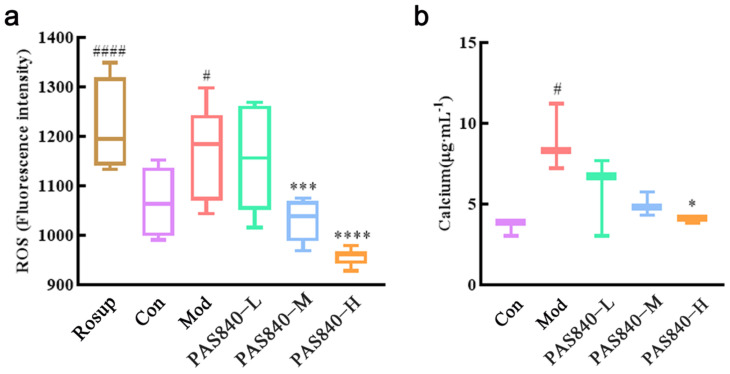
Effects of PAS840 on the content of ROS and Ca^2+^ in injured cells. (**a**) Effect of PAS840 on fluorescence intensity of ROS in injured cells. (**b**) Effect of PAS840 on Ca^2+^ in injured cells. Mean ± median. n = 6. ^#^
*p* < 0.05, ^####^
*p* < 0.0001 vs. Con group; * *p* < 0.05, *** *p* < 0.001, **** *p* < 0.0001 vs. Mod group.

**Figure 10 biology-14-00193-f010:**
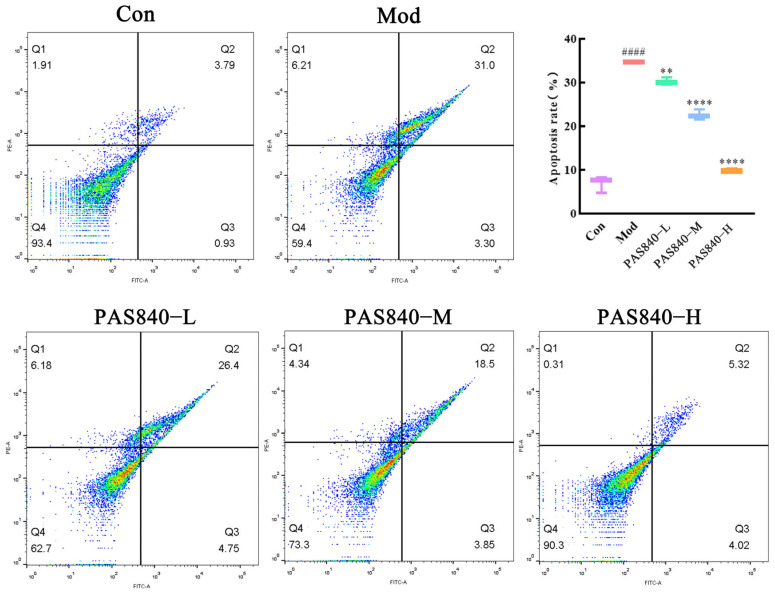
Effects of PAS840 on apoptosis rate of injured cells. Mean ± median. n = 3. ^####^
*p* < 0.0001, vs. Con group; ** *p* < 0.01, **** *p* < 0.0001 vs. Mod group.

**Figure 11 biology-14-00193-f011:**
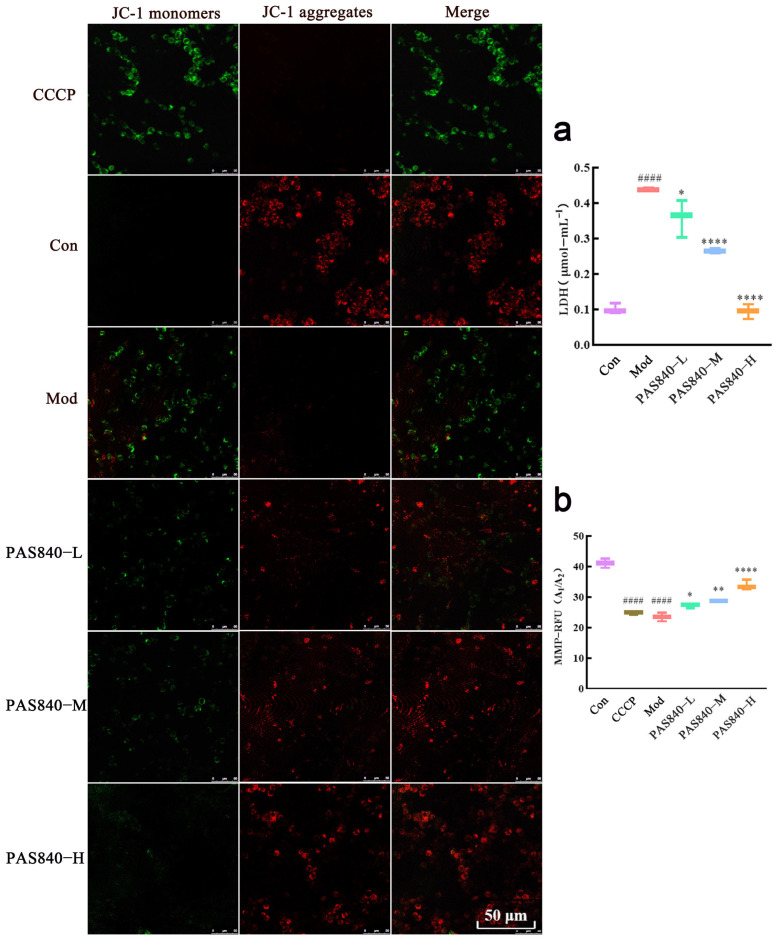
Effects of PAS840 on apoptosis and MMP of injured cells. (**a**) Apoptosis rate. (**b**) The expression rate of MMP. Mean ± median. n = 3. ^####^
*p* < 0.0001 vs. Con group; * *p* < 0.05, ** *p* < 0.01, **** *p* < 0.0001 vs. Mod group.

**Figure 12 biology-14-00193-f012:**
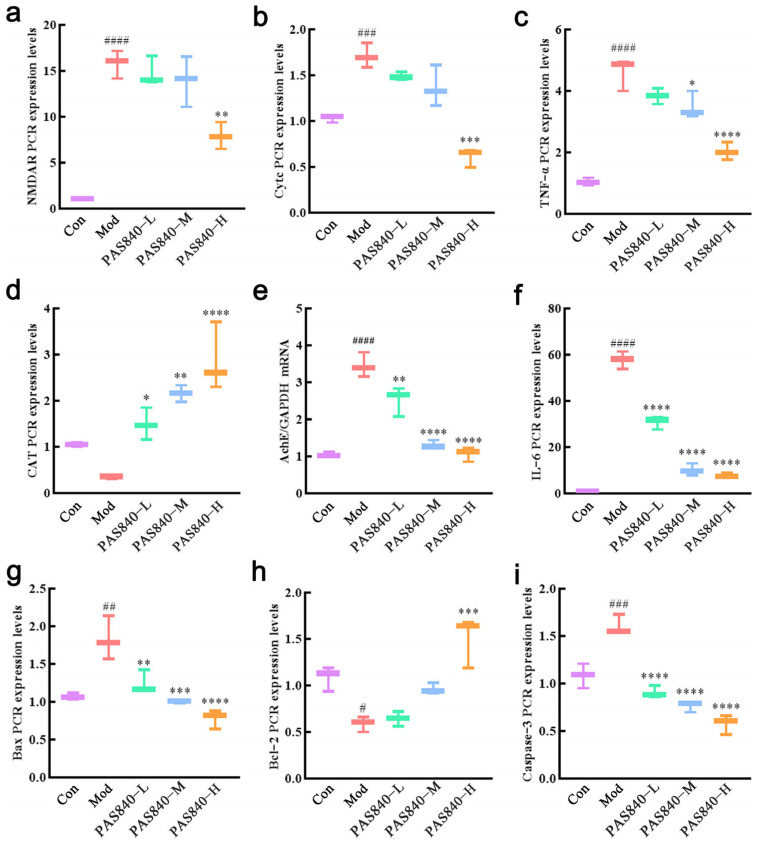
Effects of PAS840 on mRNA expression. (**a**) The mRNA expression of NMDAR1. (**b**) The mRNA expression of Cytc. (**c**) The mRNA expression of TNF-α. (**d**) The mRNA expression of CAT. (**e**) The mRNA expression of AchE. (**f**) The mRNA expression of IL-6. (**g**) The mRNA expression of Bax. (**h**) The mRNA expression of Bcl-2. (**i**) The mRNA expression of Caspase-3. Mean ± median. n = 3. ^#^
*p* < 0.05, ^##^
*p* < 0.01 ^###^
*p* < 0.001, ^####^
*p* < 0.0001 vs. Con group; * *p* < 0.05, ** *p* < 0.01, *** *p* < 0.001, **** *p* < 0.0001 vs. Mod group.

**Figure 13 biology-14-00193-f013:**
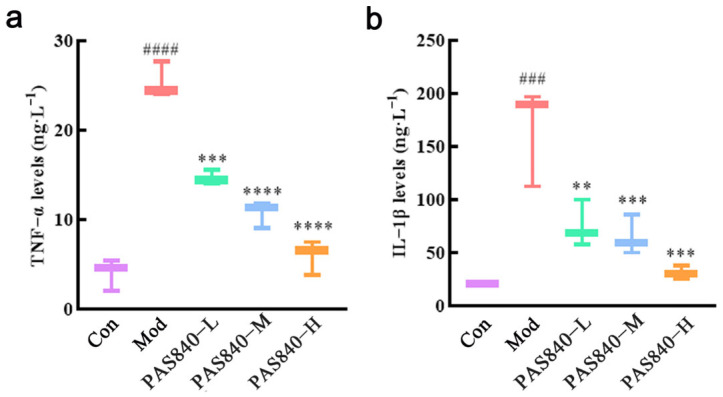
Effect of PAS840 on expression of TNF-α and IL-1β in culture medium of injured cells. (**a**) The expression level of TNF-α. (**b**) The expression level of IL-1β. Mean ± median. n = 3. ^###^
*p* < 0.001, ^####^
*p* < 0.0001 vs. Con group; ** *p* < 0.01, *** *p* < 0.001, **** *p* < 0.0001 vs. Mod group.

**Figure 14 biology-14-00193-f014:**
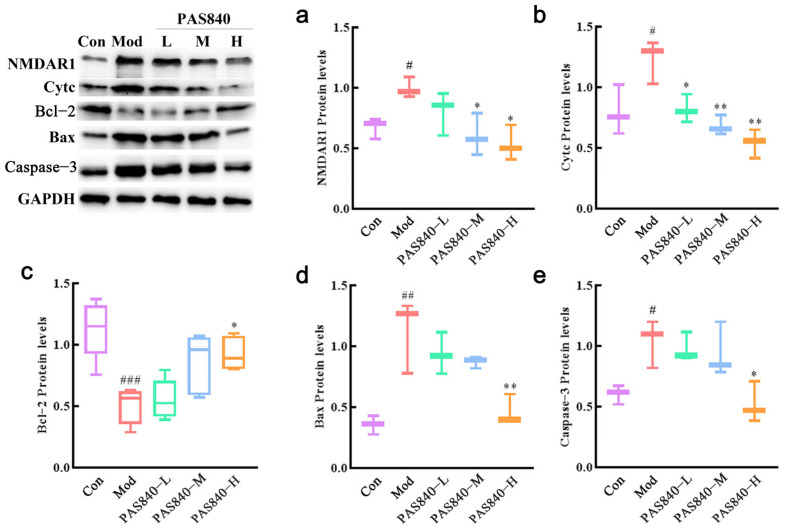
Effects of different concentrations of PAS840 on the expression of NMDAR1, Cytc, Bcl-2, Bax, and Caspase-3 proteins in injured cells. (**a**) The expression level of NMDAR1. (**b**) The expression level of Cytc. (**c**) The expression level of Bcl-2. (**d**) The expression level of Bax. (**e**) The expression level of Caspase-3. Mean ± median. n = 3. ^#^
*p* < 0.05, ^##^
*p* < 0.01 ^###^
*p* < 0.001 vs. Con group; * *p* < 0.05, ** *p* < 0.01 vs. Mod group.

**Table 1 biology-14-00193-t001:** PAS840 peptide chromatographic gradient.

Time (min)	Proportion of Mobile Phase B (%)
0–7	8
7–55	12
55–65	30
65–66	40
66–80	95
80	95

**Table 2 biology-14-00193-t002:** Primer sequences.

Name		Primer Sequence (5′-3′)
GAPDH	Forward	GGGGCTCTCTGCTCCTCCCTGF
	Reverse	CGGCCAAATCCGTTCACACCG
NMDAR1	Forward	GGACTGACTACCCGAATGTCCA
	Reverse	GTAGACTCGCATCATCTCAAACCA
Cytc	Forward	CCTTTGTGGTGTTGACCAGC
	Reverse	CCATGGAGGTTTGGTCCAGT
Bcl-2	Forward	GGACTGGGTGAGAAACGAGC
	Reverse	TTTCCGGCTCTTGTGGAAGC
Bax	Forward	AGCGAGACCTGGAGCAAGCC
	Reverse	GCACTGTCACCTGGAAGCAGAG
Caspase-3	Forward	TGGACTGCGGTATTGAGACA
	Reverse	GGGTGCGGTAGAGTAAGCAT
TNF-α	Forward	AAAGGACACCATGAGCACGGAAAG
	Reverse	CGCCACGAGCAGGAATGAGAAG
IL-6	Forward	GGAGAGGAGACTTCACAGAGGA
	Reverse	ACTCCAGAAGACCAGAGCAGAT

**Table 3 biology-14-00193-t003:** Top 20 small molecule compounds with highest relevance.

Number	Name	ID	Degree	Structures
1	(S)-N-Methylcoclaurine	PA3	67	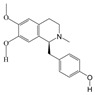
2	(S)-Reticuline	PA4	72	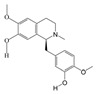
3	1,2,3-Trihydroxybenzene	PA5	5	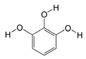
4	3-Methyl-L-tyrosine	PA17	3	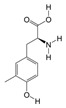
5	4-(3-Methylbut-2-enyl)-L-tryptophan	PA19	20	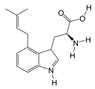
6	Capsaicin	PA30	77	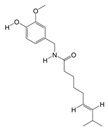
7	D-Phenylalanine	PA34	23	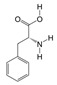
8	Deoxycholic acid	PA35	37	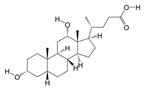
9	L-Phenylalanine	PA60	22	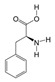
10	PGA1	PA72	84	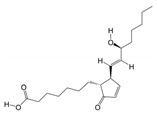

**Table 4 biology-14-00193-t004:** Peptides used for molecular docking.

Number	Peptide	ID	Degree
1	KDEGLNGFYK	peptide47	76
2	AGNALVEFPK	peptide48	72
3	LGEFEKPPPKP	peptide51	72
4	VGLEQYVPPK	peptide55	75
5	FDLVSR	peptide78	74
6	TPFYLR	peptide94	75
7	FDQLGR	peptide108	78
8	TFYNELR	peptide114	75
9	ATFDQLGR	peptide143	74
10	LFLDKLR	peptide173	81

## Data Availability

All the data are uploaded to the Mendeley Data database (https://data.mendeley.com/datasets/5b9jkrhdfs/1, accessed on 24 December 2024). The data set name is the same as the article title, DOI: 10.17632/5b9jkrhdfs.1.

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
