# Peer review of "Periplaneta americana Extract Protects Glutamate-Induced Nerve Cell Damage by Inhibiting N-Methyl-D-Aspartate Receptor Activation"

_biology, 2025, doi:10.3390/biology14020193_

Round 1

Reviewer 1 Report

Comments and Suggestions for Authors

Article SummaryThe article titled “Periplaneta americana (L.) extract alleviates the disease progression of Alzheimer's disease by inhibiting the expression of NMDAR in nerve cells” explores the neuroprotective potential of PAS840 against Alzheimer’s Disease (AD) by combining advanced methodologies such as LC-MS/MS, network pharmacology, and an in vitro excitotoxicity model using Glu-induced PC12 cells. The study is commendable for its multidisciplinary approach, effectively linking molecular, cellular, and bioinformatic insights. The discussion highlights PAS840’s capacity to modulate critical pathways, such as calcium signaling, oxidative stress, and inflammation, which are key in AD pathology. However, some areas of the manuscript would benefit from further refinement, particularly in the contextualization of findings, clarity of language, and integration of experimental data with literature.

ReviewThe review of the topic addressed in the article is well-structured and makes a valuable contribution to the field of neurodegenerative disease research, particularly in the context of Alzheimer's Disease (AD). The study on the neuroprotective effects of PAS840 provides a robust exploration of the molecular mechanisms underlying excitatory neurotoxicity, addressing a clear gap in understanding the role of bioactive compounds in AD prevention and treatment. The references are well-chosen, and most cited studies complement the findings effectively. However, some weaknesses include methodological imprecision and lack of clarity in the theoretical methods, which may affect a better understanding of the approaches used. The absence of sufficient detail regarding the parameters of the computational tools also limits the reproducibility of the results.

The manuscript titled “Periplaneta americana (L.) extract alleviates the disease progression of Alzheimer's disease by inhibiting the expression of NMDAR in nerve cells” presents significant findings. However, to strengthen the data and ensure clarity, the suggested revisions should be fully addressed.

Major revisions

A complete revision of the manuscript is necessary as there is "excessive repetition" of information in all sections.

Carefully check all the ions mentioned and use the correct nomenclature. Change the forms: “Mg2+, Ca2+, and CO2” to “Mg²⁺, Ca²⁺, and CO2”.

Line 177:

The authors used the SwissADME server. This server only predicts administration, distribution, metabolism and excretion. Why were servers that predict toxicity not used?

Line 179:

Please specify the “screening” method used. Was it a virtual screening method? What software was used?

Line 182:

In the sentence: “To obtain the related protein information, the peptide sequences that scored highest”, method used. Was it a virtual screening method? What software was used?

Lines 188 and 189:

When was the NoverPor tool (https://www.novoprolab.com/tools/convert-peptide-to-smiles-string) used? It seems to be currently unavailable or the address given is incorrect. Please check this.

Lines 229 and 238:

2.4.7. Molecular docking: Were the simulations rigid or flexible? Was the docking method blind or site-directed? If the docking was site-directed, please include the box sizes of all the simulations calculated, as well as the reaction centers. If a "seed" was used in the "conf" file, please clarify this in the manuscript. For suggestions on this topic, authors can consult: 10.1007/s13205-022-03406-w, 10.1016/j.advms.2023.10.002, or 10.3390/pr12050885.

Lines 247 to 252 - Rewrite the sentence for clarity:

“(Blank group) add 50 μL of special medium, continue to incubate for 24 h, observe the cell morphology under an inverted microscope and take pictures to record, add 10 μL of CCK-8 to each well, and after 1 h of reaction, use a multi-functional enzyme labeling instrument to detect the absorbance (OD) value of each group at 450 nm, calculate the survival rate of the cells (%), and determine the safe dosage of PAS840.”

 Lines 256 to 261 - Rewrite the sentence for clarity:

The authors repeat information that has already been said... I suggest condensing the sentences to make the reading more fluid.

Lines 281 to 282: What do the authors mean by: "Record data and analyze it."?

Lines 284 to 286 - Rewrite the sentence for clarity:

DCFH-DA probe was used to stimulate the cells to enter the cell membrane and interact with intracellular ROS to generate strong fluorescent DCF, and the fluorescence intensity was positively correlated with the ROS production.

Lines 284 to 286 - Rewrite the sentence for clarity:

Items (2.11, 2.12, and 2.14) appear to be step-by-step instructions from a laboratory protocol. Therefore, please rewrite them following the standard format for methods descriptions in scientific manuscripts."

Line 318 1× binding solution”. What solution are the authors referring to? Please clarify.

Lines 320 and 321: single-positive and double-positive tubes” Please clarify.

Line 434: Figure 3” I suggest adding Figure 3B below 3A for better visualization. It is also recommended that authors maximize the size of all images to make them easier to view.

Line 438: PA+numbers” Please clarify.

Line 454 to 452: Add all Gibbs free energies (∆G) obtained from docking simulations of 10 best ligands and peptides, respectively. I also suggest that the authors add the list of amino acids of the proteins/receptors used in the simulations that maintained interactions with the ligands or peptides.

Line 489: Con group” Please clarify.

Line 509: Mod group” Please clarify.

Minor revisions

Line 47 - Rewrite the sentence for clarity:

“NMDAR” to “NMDAR (N-methyl-D-aspartate receptor or NMDA receptor)”

 Lines 52 to 56 - Rewrite the sentence for clarity:

“During normal excitability, after NMDAR is activated by Glutamic acid (Glu), the main excitatory neurotransmitter in neuronal cells of the brain, the postsynaptic neuron undergoes a depolarization reaction in which Mg2+ ions leave the channel, and Ca2+ ions flow into the channel one after another, a process that has a farreaching and important, wide-ranging effect on learning and memory functions.” to “During normal excitability, after the NMDAR is positive modulator by glutamic acid (Glu), the main excitatory neurotransmitter in neuronal cells of the brain, the postsynaptic neuron undergoes a depolarization process in which Mg²⁺ ions blocking the channel are removed, allowing Ca²⁺ ions to flow through the channel and enter the nerve cell. This process has a far-reaching and significant impact on learning and memory functions.”

Lines 56 to 58: I suggest deleting the sentence: “However, if glutamate is released in excess, it stimulates neuronal death in the mammalian central nervous system, i.e., it manifests as excitatory neurotoxicity [4]” as since the following sentence presents the same information in different words.

Line 58 - Rewrite the sentence for clarity:

“over-activate” to “overactivate”

Lines 58 to 64: Sentences that are too long make it difficult to fully understand the information presented. I suggest rewriting into shorter sentences to make the information presented clearer and more objective.

Line 72: Delete the expression: “genus Periplaneta”. When the authors mention the name of the species (Periplaneta americana), the genus is already implied.

Line 78: Write the meaning of "PAS840".

Line 82 - Rewrite the sentence for clarity:

“PC12 cells.” to “PC12 cells (Cell line derived from a rat pheochromocytoma, a tumour of the adrenal medulla).”

Line 111 - Rewrite the sentence for clarity:

Periplaneta americana (L.)” toP. americana (L.)”

Line 113 - Rewrite the sentence for clarity:

Periplaneta americana (L.)” toP. americana

Line 116: “Water”. Specify the type of water used.

Line 119: Delete the expression: “cell experiments.”

Line 120: I suggest isolating item 2.4.1, as this concerns experimental data rather than theoretical data. Furthermore, I recommend replacing “PAS840 substance composition identification” with “Mass spectrometry identification of compounds present in PAS840”. In this case, it would become: 2.4 Mass spectrometry identification of compounds present in PAS840.

I strongly suggest using the term “Network pharmacology” instead of “Cyberpharmacological”. Also, please make it clear what “IS” means. In this case, it would become: 2.5.

 Lines 122 to 128 - Rewrite the sentence for clarity:

“To detect small-molecule compounds in the PAS840 samples were gathered and analyzed using Liquid chromatography-MS/MS(LC-MS/MS), then weighed and vortexed for 30 s in 600 µL of methanol containing 4-chloro-L-phenylalanine (4 ppm). A tissue grinder (MB-96; USA Wall) was used to ground the samples for 120 s at 50 Hz. Subsequently, the samples were centrifuged for 10 min at 12,000 rpm after sonication at 24 ± 2 °C. The supernatant was filtered through a 0.22 μm filter membrane. Finally, the filtrate was introduced into the test bottle for LC-MS analysis.” to “The PAS840 samples were gathered and ground using a tissue grinder (MB-96; USA Wall) for 120 seconds at 50 Hz. After grinding, the samples were vortexed for 30 seconds in 600 µL of methanol containing 4-chloro-L-phenylalanine (4 ppm) to facilitate extraction. The samples were then centrifuged for 10 minutes at 12,000 rpm following sonication at 24 ± 2°C. The supernatant was filtered through a 0.22 μm filter membrane, and the filtrate was finally introduced into the test bottle for LC-MS/MS analysis to detect small-molecule compounds.”

Line 198 - Rewrite the term for:

“three” to “four”

Line 202 - Rewrite the term for:

“disease” to “Disease”

Line 239:

I strongly suggest that the topic "2.5. Cell lines" be deleted entirely.

 Line 243 - Rewrite the sentence for clarity:

“PC12 cells …” to “PC12 cells (purchased from Wuhan Punosai Life Science and Technology Company Limited - Batch No. 8300L222009) ...”

 Line 363 - Rewrite the term for:

“3.8” to “2.8”

Line 366 - Rewrite the sentence for clarity:

“BCA method” to “BCA (Bicinchoninic Acid Assay) method”

Lines 393 and 397:

Change the term: “Wayne diagram” to “Venn diagram”.

Line 590 - Rewrite the sentence for:

“in vivo” toin vivo”. In addition, if there are others, change them to the correct form.

Lines 595 to 596 - Rewrite the sentence for:

“in vitro” to “in vitro”. In addition, if there are others, change them to the correct form.

 Lines 602, 605 and 608 - Rewrite the sentence for clarity:

“PA” toP. americana”. In addition, if there are others, change them to the correct form.

 Line 700:

Delete the expression:This section is mandatory, with one or two paragraphs to end the main text.”

Comments on the Quality of English Language

It is recommended that the manuscript is proofread by a specialist company or by a native professional of the English language.

Author Response

We are sincerely grateful to the reviewer for conducting such a meticulous and comprehensive review of our manuscript and for providing us with so many precious and valuable suggestions. Your feedback has made us aware of the various shortcomings in our manuscript writing, and we feel quite embarrassed about our less-than-ideal use of English. Meanwhile, taking into account the suggestions from other reviewers, we have revised both the title and the theme of our manuscript, and made adjustments to the entire text. We would be very grateful if you could take a moment to review our newly uploaded manuscript and provide us with your new insights, which will surely assist us in further refining the manuscript.

Additionally, in response to your previous questions, we have made corresponding answers or corrections below:

Major revisions

Comments 1: A complete revision of the manuscript is necessary as there is "excessive repetition" of information in all sections.

Carefully check all the ions mentioned and use the correct nomenclature. Change the forms: “Mg2+, Ca2+, and CO2” to “Mg²⁺, Ca²⁺, and CO2”.

Response 1: We are very grateful that you have pointed out our clumsy and careless mistakes. We have re-examined the entire text and checked and revised them one by one.

Comments 2: Line 177: The authors used the SwissADME server. This server only predicts administration, distribution, metabolism and excretion. Why were servers that predict toxicity not used?

Response 2: In the network pharmacology experimental steps we learned previously, we learned from a more professional senior colleague about predicting drug targets in the SwissADME database. We indeed did not conduct toxicity prediction and analysis as professionally and carefully as you. However, in subsection 3.7 of the experiment, we further examined the drug's toxic effects through cell experiments. I believe that the results of this experiment are more convincing regarding the drug's toxic effects than the prediction results.

Comments 3: Line 179: Please specify the “screening” method used. Was it a virtual screening method? What software was used?

Response 3: Thank you for your question. We downloaded the predicted target information of each compound from the SwissADME database. There is an indicator in the table named "probability", which represents the likelihood of the credibility of this target. Therefore, we excluded all targets with a probability < 0.01 to increase the prediction credibility. The detailed tables downloaded for each compound have been recorded in the raw data and made public in the Mendeley Data database.

Comments 4: Line 182: In the sentence: “To obtain the related protein information, the peptide sequences that scored highest”, method used. Was it a virtual screening method? What software was used?

Response 4: Regarding this screening method, we sorted the data in ascending order based on the -10 lgP value of each polypeptide. The -10 lgP value is the condition provided by the commissioning company that completed the polypeptide omics sequencing for us to screen the most reliable polypeptides in the Database Peptides file.

Comments 5: Lines 188 and 189: When was the NoverPor tool (https://www.novoprolab.com/tools/convert-peptide-to-smiles-string) used? It seems to be currently unavailable or the address given is incorrect. Please check this.

Response 5: We have tested the login on this website and found that the website address is correct. It is possible that the website was under maintenance or there was network fluctuation when you logged in. We hope you can try to access it again.

Comments 6: 2.4.7. Molecular docking: Were the simulations rigid or flexible? Was the docking method blind or site-directed? If the docking was site-directed, please include the box sizes of all the simulations calculated, as well as the reaction centers. If a "seed" was used in the "conf" file, please clarify this in the manuscript. For suggestions on this topic, authors can consult: 10.1007/s13205-022-03406-w, 10.1016/j.advms.2023.10.002, or 10.3390/pr12050885.

Response 6: Thank you to the reviewer for the suggestions regarding our molecular docking experiment. We have supplemented the docking method on line 239 in the new version of the manuscript. Among them, the search space volume established by our docking is > 27000 Angstrom³ (see FAQ), and the random seed file used has also been uploaded in the supplementary file. In addition, we have taken screenshots of all the conditions used in each docking.

Comments 7: Lines 247 to 252 - Rewrite the sentence for clarity: “(Blank group) add 50 μL of special medium, continue to incubate for 24 h, observe the cell morphology under an inverted microscope and take pictures to record, add 10 μL of CCK-8 to each well, and after 1 h of reaction, use a multi-functional enzyme labeling instrument to detect the absorbance (OD) value of each group at 450 nm, calculate the survival rate of the cells (%), and determine the safe dosage of PAS840.”

Response 7: Thank you to the reviewer for questioning our language logic. We have also rewritten all the sentences from lines 244 to 268, And according to the suggestions of other reviewers, the Blank group has been renamed the Naive group.

Comments 8: Lines 256 to 261 - Rewrite the sentence for clarity: The authors repeat information that has already been said... I suggest condensing the sentences to make the reading more fluid.

Response 8: We have rewritten the entire subsection and streamlined its content.

Comments 9: Lines 281 to 282: What do the authors mean by: "Record data and analyze it."?

Response 9: We originally intended to express the recording of the OD value and the calculation of the corresponding cytokine content. Thank you for your reminder. We have already made corrections and supplements in lines 277-288 of the new manuscript. The test instruction manual of the kit has been uploaded to the supplementary file.

Comments 10: Rewrite the sentence for clarity: DCFH-DA probe was used to stimulate the cells to enter the cell membrane and interact with intracellular ROS to generate strong fluorescent DCF, and the fluorescence intensity was positively correlated with the ROS production.

Response 10: We have rewritten the sentence. For details, please refer to lines 288-291 of the new manuscript.

Comments 11: Lines 284 to 286 - Rewrite the sentence for clarity: Items (2.11, 2.12, and 2.14) appear to be step-by-step instructions from a laboratory protocol. Therefore, please rewrite them following the standard format for methods descriptions in scientific manuscripts."

Response 11: Thank you to the reviewer for the query. All of our experimental steps were carried out in accordance with the kit instructions mentioned in the relevant subsections. Meanwhile, in order to avoid repetitive operations in the content of this subsection, we have provided a streamlined description. But we have also added annotations in three subsections, and readers can obtain detailed operation manuals in the supplementary file.

Comments 12: Line 318: “1× binding solution”. What solution are the authors referring to? Please clarify. Lines 320 and 321: “single-positive and double-positive tubes” Please clarify.

Response 12:

The 1x binding solution is a working solution that has already been prepared by the manufacturer in the apoptosis detection kit and is a fully functional working solution that can be directly used for experimental operations. For detailed instructions, please refer to the manual in the supplementary materials.

  1. Single Positive Tube

In flow cytometry, a single positive tube generally refers to the sample tube where the cell population that expresses only one fluorescent marker is located in a multicolor fluorescent labeling experiment.

For example, when we stain cells with multiple fluorescently labeled antibodies (such as fluorochrome-labeled antibodies A, B, and C), in a single positive tube, we may obtain a cell population that shows a positive reaction only to antibody A. That is, this cell population only expresses the antigen corresponding to antibody A and does not express the antigens corresponding to antibodies B and C. By using a single positive tube, it can help us determine the individual positive signal range and fluorescence intensity distribution of each fluorescently labeled antibody, which is crucial for setting correct compensation in subsequent multicolor fluorescence analysis. Because the spectra emitted by different fluorochromes may overlap to some extent, in order to accurately distinguish cell populations labeled with different fluorochromes, we need to adjust the compensation settings of the instrument based on the data from the single positive tube so that different fluorescent signals can be accurately identified and distinguished.

  1. Double Positive Tube

A double positive tube refers to the sample tube where the cell population that simultaneously expresses two fluorescent markers is located in a multicolor fluorescent labeling experiment.

Continuing with the above example, if we find a cell population in a sample tube that shows a positive reaction to both antibody A and antibody B, but a negative reaction to antibody C, then this sample tube is a double positive tube for A and B. The double positive tube is very useful for studying the co-expression of two antigens on the cell surface or within the cell. For instance, in immunological research, we may be interested in whether a certain lymphocyte simultaneously expresses two surface markers, CD4 and CD25. Then, we can use a double positive tube to detect the cell population that expresses both CD4 and CD25. Through the double positive tube, we can determine the proportion, fluorescence intensity, and other information of cells that express two specific antigens simultaneously, providing an important basis for our in-depth understanding of cell phenotypes and functions.

Comments 13: Line 434: “Figure 3” I suggest adding Figure 3B below 3A for better visualization. It is also recommended that authors maximize the size of all images to make them easier to view.

Response 13: We have already rectified Figure 3 to make its content more easily visible.

Comments 14: Line 438: “PA+numbers” Please clarify. 

Response 14: First, we sorted all compounds in PAS840 by the first letter of their names. The ID names were named PA + sorting number based on the sorting results. This content has also been supplemented on lines 449-450 of the manuscript to facilitate readers' understanding.

Comments 15: Line 454 to 452: Add all Gibbs free energies (∆G) obtained from docking simulations of 10 best ligands and peptides, respectively. I also suggest that the authors add the list of amino acids of the proteins/receptors used in the simulations that maintained interactions with the ligands or peptides.

Response 15: We have shown the corresponding list and docking scores of all peptides and small molecules against target proteins in Figure 5, which I believe should clearly represent the binding of our drugs to proteins.

Comments 16: Line 489: “Con group” Please clarify. Line 509: “Mod group” Please clarify.

Response 16: We have provided annotations for the Naive group, Con group, and Mod group in line 490 and lines 511-514 of the new manuscript.

Minor revisions

Comments 1: Line 47 - Rewrite the sentence for clarity:

“NMDAR” to “NMDAR (N-methyl-D-aspartate receptor or NMDA receptor)”

Response 1: We have already rectified it and made supplements in the title, abstract, and on line 53 of the new manuscript.

Comments 2: Lines 52 to 56 - Rewrite the sentence for clarity:

“During normal excitability, after NMDAR is activated by Glutamic acid (Glu), the main excitatory neurotransmitter in neuronal cells of the brain, the postsynaptic neuron undergoes a depolarization reaction in which Mg2+ ions leave the channel, and Ca2+ ions flow into the channel one after another, a process that has a farreaching and important, wide-ranging effect on learning and memory functions.” to “During normal excitability, after the NMDAR is positive modulator by glutamic acid (Glu), the main excitatory neurotransmitter in neuronal cells of the brain, the postsynaptic neuron undergoes a depolarization process in which Mg²⁺ ions blocking the channel are removed, allowing Ca²⁺ ions to flow through the channel and enter the nerve cell. This process has a far-reaching and significant impact on learning and memory functions.”

Response 2: Thank you to the reviewer for the modification suggestions on our originally clumsy language. We have already made rectifications.

Comments 3: Lines 56 to 58: I suggest deleting the sentence: “However, if glutamate is released in excess, it stimulates neuronal death in the mammalian central nervous system, i.e., it manifests as excitatory neurotoxicity [4]” as since the following sentence presents the same information in different words.

Response 3: Thank you for your opinion. However, we still hope to retain this sentence. It can serve as a link between the preceding and following parts, enabling readers to immediately understand the central idea that needs to be highlighted in our lengthy mechanism explanation. So, once again, thank you for your suggestion, and we are truly sorry about this.

Comments 4: Line 58 : Rewrite the sentence for clarity: “over-activate” to “overactivate”

Response 4: Thank you for the reviewer's suggestions. We have already made corrections at line 64 of the new manuscript.

Comments 5: Lines 58 to 64: Sentences that are too long make it difficult to fully understand the information presented. I suggest rewriting into shorter sentences to make the information presented clearer and more objective.

Response 5: Thank you for the reviewer's suggestions. This process is a complete and complex pathogenesis process, and it is very difficult for us to further streamline it, as it may lead to misleading information and cause readers to misunderstand. Therefore, the content that you asked us to cut in Comments 3 is precisely a summary of this, which aims to prevent readers from making mistakes in understanding.

Comments 6: Line 72: Delete the expression: “genus Periplaneta”. When the authors mention the name of the species (Periplaneta americana), the genus is already implied.

Response 6: Thank you for the reviewer's suggestions. We have already made corrections at line 77 in the new manuscript.

Comments 7: Line 78: Write the meaning of "PAS840".

Response 7: The name of PAS840 is named based on the process characteristics mentioned in subsection 2.3, where adsorption is carried out using an S-8 macroporous resin column and extraction is performed with 40% ethanol. At the same time, we have also made rectifications at lines 83-85 of the new manuscript.

Comments 8: Line 82 - Rewrite the sentence for clarity: “PC12 cells.” to “PC12 cells (Cell line derived from a rat pheochromocytoma, a tumour of the adrenal medulla).”

Response 8: Thank you for the reviewer's suggestions. We have already made corrections at line 88 in the new manuscript.

Comments 9: Line 111 - Rewrite the sentence for clarity: “Periplaneta americana (L.)” to “P. americana (L.)”. Line 113 - Rewrite the sentence for clarity: “Periplaneta americana (L.)” to “P. americana”

Response 9: Thank you for the reviewer's suggestions. We have also made rectifications to similar parts throughout the entire text.

Comments 10: Line 116: “Water”. Specify the type of water used.

Response 10: The “Water” is ultrapure water,.and we have also made rectifications at lines 127 of the new manuscript. 

Comments 11: Line 119: Delete the expression: “cell experiments.”

Response 11: Thank you for the reviewer's suggestions. We have already made corrections at line 129-130 in the new manuscript.

Comments 12: Line 120: I suggest isolating item 2.4.1, as this concerns experimental data rather than theoretical data. Furthermore, I recommend replacing “PAS840 substance composition identification” with “Mass spectrometry identification of compounds present in PAS840”. In this case, it would become: 2.4 Mass spectrometry identification of compounds present in PAS840.

Response 12: Thank you for the reviewer's proposal. We have already isolated and modified the rectified subsection.

Comments 13: I strongly suggest using the term “Network pharmacology” instead of “Cyberpharmacological”. Also, please make it clear what “IS” means. In this case, it would become: 2.5.

Response 13: We appreciate your pointing out our inappropriate use of words in this subsection. We have already made modifications. IS is another brain disorder, namely ischemic stroke, that we are researching. It was mistakenly included during the writing process, and we have already made rectifications.

Comments 13: Lines 122 to 128 - Rewrite the sentence for clarity:

“To detect small-molecule compounds in the PAS840 samples were gathered and analyzed using Liquid chromatography-MS/MS(LC-MS/MS), then weighed and vortexed for 30 s in 600 µL of methanol containing 4-chloro-L-phenylalanine (4 ppm). A tissue grinder (MB-96; USA Wall) was used to ground the samples for 120 s at 50 Hz. Subsequently, the samples were centrifuged for 10 min at 12,000 rpm after sonication at 24 ± 2 °C. The supernatant was filtered through a 0.22 μm filter membrane. Finally, the filtrate was introduced into the test bottle for LC-MS analysis.” to “The PAS840 samples were gathered and ground using a tissue grinder (MB-96; USA Wall) for 120 seconds at 50 Hz. After grinding, the samples were vortexed for 30 seconds in 600 µL of methanol containing 4-chloro-L-phenylalanine (4 ppm) to facilitate extraction. The samples were then centrifuged for 10 minutes at 12,000 rpm following sonication at 24 ± 2°C. The supernatant was filtered through a 0.22 μm filter membrane, and the filtrate was finally introduced into the test bottle for LC-MS/MS analysis to detect small-molecule compounds.”

Response 13: Thank you for your opinion. We have rewritten it and streamlined the entire paragraph to a certain extent.

Comments 14: Line 198 - Rewrite the term for: “three” to “four”

Response 14: Thank you for your opinion. We found that we repeated the description of a database, and in fact, it should be written as "Three".

Comments 15: Line 202 - Rewrite the term for: “disease” to “Disease”

Response 15: Thank you for your opinion. We have made corresponding rectifications accordingly.

Comments 16: Line 239: I strongly suggest that the topic "2.5. Cell lines" be deleted entirely.

Response 16: Thank you for your opinion. We have decided to delete this subsection.

Comments 17: Line 243 - Rewrite the sentence for clarity: “PC12 cells …” to “PC12 cells (purchased from Wuhan Punosai Life Science and Technology Company Limited - Batch No. 8300L222009) ...”

Response 17: Thank you for your opinion. At the same time, we have also combined the requirements of other reviewers and moved it into subsection 2.2.

Comments 18: Line 363 - Rewrite the term for: “3.8” to “2.8”

Response 18: Thank you for pointing out our writing mistakes. We have already made rectifications.

Comments 19: Line 366 -Rewrite the sentence for clarity: “BCA method” to “BCA (Bicinchoninic Acid Assay) method”

Response 19: Thank you for your opinion. We have made modifications at line 378 in the new manuscript.

Comments 20: Lines 393 and 397: Change the term: “Wayne diagram” to “Venn diagram”.

Response 20: Thank you for your opinion. We have made modifications at line 405 and 409 in the new manuscript.

Comments 21: Line 590 - Rewrite the sentence for: “in vivo” to “in vivo”. In addition, if there are others, change them to the correct form.

Response 21: Thank you for pointing out our mistakes. We have already made unified modifications to similar problems throughout the entire text.

Comments 22:  Lines 602, 605 and 608 - Rewrite the sentence for clarity:“PA” to “P. americana”. In addition, if there are others, change them to the correct form.

Response 22: Thank you for pointing out our mistakes. We have already made unified modifications to similar problems throughout the entire text.

Comments 23:  Line 700: Delete the expression: “This section is mandatory, with one or two paragraphs to end the main text.”

Response 23: Thank you for pointing out the mistakes. We have already deleted the text that comes with this template.

Reviewer 2 Report

Comments and Suggestions for Authors

This study suggests that Periplaneta americana extract (PAS840) targets multiple pathways in Alzheimer’s disease (AD) like reducing oxidative stress, inflammation, and apoptosis in glutamate-injured cells. By modulating NMDAR1 activity and improving cell viability, PAS840 demonstrates potential as a multi-target therapeutic agent for slowing AD progression. Here are some suggestions below:

  • Upon reviewing this manuscript, it appears that the authors may have submitted an incorrect or incomplete version of the manuscript. Before submission, it is crucial to ensure that all results are properly included and the manuscript is properly reviewed.
  • The authors should provide a clear rationale or reference for the selected dose of PAS840. Adding this information should enhance the understanding of author.
  • In the methods section, while the measurement of SOD, GSH, and MDA in treated cells is described, it lacks sufficient detail. Including a comprehensive description of the experimental procedures and citing appropriate references should improve clarity.
  • For the calcium assay, the authors should provide details on how calcium levels were measured, including any dyes, kits, or reagents used. Providing these specifics, along with references or detailed protocols if methods were developed by own, would greatly enhance the representation of results. The authors should carefully revise and rewrite the all the methods section for better clarity and accuracy.
  • Advanced techniques such as LC-MS/MS and network pharmacology are well used. However, the manuscript should be better from additional details about the molecular docking process, such as scoring metrics and validation procedures, to ensure transparency and reproducibility.
  • In the results section, scale bars should be added to all relevant images to improve data representation and interpretation.
  • While the study highlights PAS840’s neuroprotective role, the authors should provide a stronger justification for selecting PAS840 over other candidate compounds. This should highlight its significance and relevance to the field.
  • The manuscript provides a solid background on excitatory neurotoxicity and glutamate-induced damage. However, a more critical comparison of PAS840 with existing neuroprotective agents targeting NMDAR1 would strengthen the context and discussion.
  • Overall, the manuscript is well-designed but lacks effective writing and representation. With improvements in structure, detail, and clarity, it has the potential to add valuable insights to the field of age-related disorder research.

Comments on the Quality of English Language

should improve.

Author Response

First of all, thank you very much for the very high evaluation you gave to our manuscript. However, at the same time, we also realize that due to our poor language skills, there are numerous mistakes and incomprehensible parts in our manuscript. We have done our utmost to integrate the opinions of four reviewers and made modifications accordingly, and we have also made significant overall changes to the entire manuscript. We hope that you can review our newly completed manuscript and help us to continue to improve the content of our manuscript.

Additionally, in response to your previous questions, we have made corresponding answers or corrections below:

Comments 1: Upon reviewing this manuscript, it appears that the authors may have submitted an incorrect or incomplete version of the manuscript. Before submission, it is crucial to ensure that all results are properly included and the manuscript is properly reviewed.

Response 1: First of all, we are very grateful for the reminder from the reviewer. We have already deeply realized our deficiencies in writing, and at the same time, we have made corresponding rectifications. We hope that the reviewer can re-examine our newly uploaded manuscript and give new opinions to help us improve this manuscript.

Comments 2: The authors should provide a clear rationale or reference for the selected dose of PAS840. Adding this information should enhance the understanding of author.

Response 2: Thank you very much for your question, reviewer. For this newly extracted novel drug, we are currently conducting tests on drug dosage and toxicity. In this manuscript, we have initially used the Cell Counting Kit-8 (CCK-8) reagent in subsection 3.7 to investigate drug toxicity and its impact on cell proliferation rate. We will select the dose that is most effective for cell proliferation and has no obvious toxicity for subsequent experiments. The CCK-8 experiment is currently the most commonly used method for exploring drug cytotoxicity and screening drug dosage [1, 2]. Of course, more experiments are needed for further verification, and this is exactly what our team plans to attempt and tackle in the later stage of the project.

Comments 3: In the methods section, while the measurement of SOD, GSH, and MDA in treated cells is described, it lacks sufficient detail. Including a comprehensive description of the experimental procedures and citing appropriate references should improve clarity.

Response 3: Thank you for your query. Indeed, in order to avoid excessive repetition in the operation of the kits, we chose to omit the relevant detailed steps. We have already provided supplementary explanations for all similar issues in subsections 2.9 - 2.13, and we have included the instruction manuals of all kits in the supplementary files so that readers can directly obtain the operation methods of the kits we used.

Comments 4: For the calcium assay, the authors should provide details on how calcium levels were measured, including any dyes, kits, or reagents used. Providing these specifics, along with references or detailed protocols if methods were developed by own, would greatly enhance the representation of results. The authors should carefully revise and rewrite the all the methods section for better clarity and accuracy.

Response 4: Thank you for your query. We have carefully examined the content of our materials section and indeed found that we did not provide information about the calcium ion detection kit. In response, we have added relevant information at line 106 of the manuscript and indicated in subsection 2.11 that readers can download the corresponding instruction manual from the supplementary files for reference.

Comments 5: Advanced techniques such as LC-MS/MS and network pharmacology are well used. However, the manuscript should be better from additional details about the molecular docking process, such as scoring metrics and validation procedures, to ensure transparency and reproducibility.

Response 5: Thank you for your suggestion. We have also received similar issues from other reviewers, so we have supplemented the content of 2.5.7. Molecular Docking. Meanwhile, in order to avoid lengthy repetition, we have explained the size of the search area in the box used for molecular docking and the use of random seeds for docking, and informed the readers that the seed files for docking can be obtained from the supplementary files and the original data publicly available in the Mendeley Data database.

Comments 6: In the results section, scale bars should be added to all relevant images to improve data representation and interpretation.

Response 6: Thank you very much for your suggestion. We have made rectifications to all the pictures in the article. We have added the relevant scale bar to the bottom right corner of the last picture of all cell pictures and enlarged the pictures to make them have higher clarity.

Comments 7: While the study highlights PAS840’s neuroprotective role, the authors should provide a stronger justification for selecting PAS840 over other candidate compounds. This should highlight its significance and relevance to the field.

Response 7: Thank you very much for the reviewer's query. As this drug is a newly discovered extract by us, there is indeed not much literature to support its pharmacological activities. This experiment is based on our team's earlier finding that PAS840 can effectively protect PC12 cells from oxidative stress damage induced by H2O2, and we have already published this finding in the Chinese Peking University Core Database. Meanwhile, we still need to continue exploring its pharmacological activities and its application prospects in neurodegenerative diseases. [3]

Comments 8: The manuscript provides a solid background on excitatory neurotoxicity and glutamate-induced damage. However, a more critical comparison of PAS840 with existing neuroprotective agents targeting NMDAR1 would strengthen the context and discussion.

Response 8: Thank you for your query. Regarding this issue, it was a mistake in our experimental design. At first, we intended to compare the drug efficacy with the most effective drugs on the market for treating Alzheimer's, but currently, there are no approved specific drugs that are clearly effective. As a result, we overlooked the possibility of comparing with NMDAR receptor antagonists. Thank you, reviewer, for your suggestion. We will add this to our team's subsequent experimental projects. Once again, thank you for your great suggestions for our experiment.

Reference:

  • [1] Chen, Z., Q. Wu and J. Wang, Antitumor effect of levetiracetam combined with temozolomide in glioma cell lines. Journal of Clinical Oncology, 2017.
  • [2] Borra, R.C., et al., A simple method to measure cell viability in proliferation and cytotoxicity assays. Brazilian Oral Research, 2009.
  • [3] Yongfang Z, Tangfei G,Peiyun X,et al., Protective effect of Periplaneta americana extract on oxidative damage of PC12 cells induced by H2O2. Chinese Journal of Hospital Pharmacy, 2023.

Reviewer 3 Report

Comments and Suggestions for Authors In this study, the authors explore the neuroprotective potential of Periplaneta americana extract (PAS840) in alleviating Alzheimer's disease (AD) by inhibiting NMDAR1 receptor activity and mitigating excitatory neurotoxicity. Through in vitro tests on glutamate-injured PC12 cells, network pharmacology, and molecular docking, PAS840 was demonstrated to enhance mitochondrial function and cell viability while lowering oxidative stress, calcium influx, inflammation, and apoptosis. The extract successfully increased anti-apoptotic Bcl-2 expression while decreasing pro-apoptotic markers (Cytc, Bax, Caspase-3) and inflammatory factors (TNF-α, IL-6). Although more in vivo research is required to validate its clinical potential, the results imply that PAS840 may be a viable natural therapy strategy for delaying AD progression. Although interesting, I have the following comments:   1. How well does the glutamate-induced PC12 cell injury model replicate the neurotoxicity and pathology seen in Alzheimer's disease? 2. Please show any dose-response figure against glutamate calibrated dose or a reference how the working concentration of PAS840 was decided. 3. Why were PC12 cells chosen over other neuronal cell lines, such as primary cortical neurons or human-induced pluripotent stem cell (iPSC)-derived neurons? 4. Use of scientific terms is highly encouraged, like imaged instead of photographed, naive instead of blank, etc. 5. Lack of in vivo data to validate the effects of PAS840. 6. My major question from this study would be how glutamate-induced toxicity can replicate the in vitro AD model without any use of amyloid beta or tau pathology analyses. 7. I believe this manuscript should be rewritten in the context of just glutamate-induced toxicity, not an AD model. 8. Comparison of these results with some neuronal cell line at least, i.e. SH-SY5Y cells or any other neurons will be highly encouraged. 9. Lines 706-708 in conclusions could be overpromising as this drug has not been well explored in neurological diseases. 10. Overall, I suggest a lot more improvement is required before this manuscript goes for publication.

Comments on the Quality of English Language

The use of scientific terms would be highly encouraged.

Author Response

We are truly very grateful to the reviewer for conducting a very careful and meticulous review of our manuscript and giving us so many precious and highly valuable suggestions. This makes us realize that there are so many deficiencies in our manuscript writing, and at the same time, we feel ashamed of our poor English wording. Meanwhile, taking into account the opinions of other reviewers, we have changed the title and the theme of the manuscript and rectified the entire text. We also realize that simple in vitro experiments do not necessarily indicate a direct connection with Alzheimer's disease (AD). It may have a broad effect on this type of disease. Therefore, we have changed the direct connection with AD to a discussion using AD as a quasi-target disease. We hope that you can take a look at our re-uploaded manuscript and also hope that you can give new suggestions to help us further improve the manuscript. Also, below is our response to the rectification of the issues you raised previously. Please have a look:

Comments 1: How well does the glutamate-induced PC12 cell injury model replicate the neurotoxicity and pathology seen in Alzheimer's disease?

Response 1: Thank you for your question, reviewer. We have already discussed Alzheimer's disease as a quasi-target disease, which can more broadly indicate that this model can be applied to neurodegenerative diseases including AD. Excitotoxicity is a pathological process caused by excessive or prolonged activation of excitatory amino acid receptors and can lead to various neurological diseases such as Alzheimer's disease. Glu has been reported as an important excitatory amino acid in the central nervous system. It exists in about one-third of the synapses in the central nervous system, can maintain synaptic stability, and plays an important role in synaptic plasticity. It also affects neuronal differentiation, migration, growth, and survival. Therefore, Glu plays an important role in neurotransmitter transmission in the early stage of Alzheimer's disease. The PC12 cell line is derived from a cloned cell line of transplanted rat adrenal pheochromocytoma. After differentiation, it has neuron-like functional characteristics and is widely used in in vitro neurobiology and research on neurological diseases. Studies have shown that NMDAR1 is coupled with calcium channels to form a receptor-ion channel complex, thereby mediating intercellular signal transduction. After Glu is induced, NMDAR1 is overactivated, resulting in a large influx of Ca2+ causing calcium overload, triggering downstream pathways, and subsequently causing neuronal damage.

Comments 2: Please show any dose-response figure against glutamate calibrated dose or a reference how the working concentration of PAS840 was decided.

Response 2: This study only conducted in vitro activity experiments on PAS840. The Cell Counting Kit-8 (CCK-8) is a commonly used experimental method in modern in vitro experiments to explore the cell proliferation rate and cytotoxic effects of drugs. The three concentrations of PAS840 in the article were selected within the concentration range where PAS840 has no significant influence on the survival rate of PC12 cells. In the new manuscript Figure 7, we used different doses of glutamate to induce cell damage and screened out the concentration of the modeling agent that can stably put the cells in the LD50 (median lethal dose) state. This concentration under the cell damage state is relatively scientific and reliable.

Comments 3: Why were PC12 cells chosen over other neuronal cell lines, such as primary cortical neurons or human-induced pluripotent stem cell (iPSC)-derived neurons?

Response 3: PC12 cells are differentiated cell lines of rat adrenal medullary pheochromocytoma and possess the general characteristics of neuroendocrine cells. Due to their passable characteristics, PC12 cells are widely used in neurophysiological and neuropharmacological research. They are commonly used in the study of neuronal differentiation and the molecular mechanisms of NGF action, and are also used to study the mechanisms by which growth factors regulate changes in gene expression in nerve cells. First of all, considering the cost and the survival rate of the cells, and secondly, combining the purpose of the research and the characteristics of PC12 cells themselves, PC12 cells were finally selected as the cells for in vitro research. Meanwhile, as an immortalized cell line, the state of this cell remains relatively stable after multiple passages, which can better improve the experimental repeatability and reduce the errors caused by internal factors.

Comments 4: Use of scientific terms is highly encouraged, like imaged instead of photographed, naive instead of blank, etc.

Response 4: Thank you for your suggestions. We deeply recognize our lack of language proficiency. We have already modified similar issues in the article. Meanwhile, we sincerely request you to review the new manuscript again and help us improve the language usage of the new manuscript, making it more complete and easier to read.

Comments 5: Lack of in vivo data to validate the effects of PAS840.

Response 5: Thank you for your query. This experiment serves as a preliminary verification for subsequent validation by the team and aims to fully demonstrate the certain effectiveness of our PAS840. Meanwhile, in order to comply with the World Animal Welfare Guidelines, we have decided to conduct in vivo experiments only after obtaining sufficient data and obtaining peer review approval, which indicates that PAS840 is effective for neurological diseases. Now, we have already obtained certain experimental data and plan to complete the animal experiments and publish the results this year.

Comments 6: My major question from this study would be how glutamate-induced toxicity can replicate the in vitro AD model without any use of amyloid beta or tau pathology analyses.

Response 6: Really, thank you very much for the questions you raised. Excitotoxicity is a pathological process caused by excessive or prolonged activation of excitatory amino acid receptors, which can lead to various neurological diseases such as Alzheimer's disease. Glu can induce overactivation of NMDAR1, resulting in a large influx of Ca2+ causing calcium overload, triggering downstream pathways, and subsequently causing neuronal damage. Neuronal damage is one of the pathological features of Alzheimer's disease. In this experiment, excessive Glu was administered to stimulate PC12 cells to simulate the increased release of Glu from the presynaptic membrane of neurons, the activation of NMDAR1, and the influx of Ca2+, thereby increasing the cell apoptosis rate and simulating the pathological features of neuronal damage after the occurrence of the disease. However, we also realize that this can broadly represent the characteristics of multiple neurodegenerative diseases. Therefore, we have changed the title and theme of the manuscript, and we hope that you can give us more suggestions to help us improve this manuscript.

Comments 7: I believe this manuscript should be rewritten in the context of just glutamate-induced toxicity, not an AD model.

Response 7: Thank you very much for your suggestion. We have already modified the title, abstract, and discussion of the article, and changed the theme to cytotoxic damage caused by NMDAR activation induced by glutamate.

Comments 8: Comparison of these results with some neuronal cell line at least, i.e. SH-SY5Y cells or any other neurons will be highly encouraged.

Response 8: Thank you very much for your suggestion. Our experiment, which serves as the foundation for subsequent in vivo experiments, indeed needs further improvement as it only simulates the neurotoxicity model through PC12 cells. We will continue to conduct in-depth exploration according to your opinion, and while carrying out in vitro experiments, we will also use other neural cell lines for in vivo experimental verification in order to obtain more powerful evidence to prove the effectiveness of our drug.

Comments 9: Lines 706-708 in conclusions could be overpromising as this drug has not been well explored in neurological diseases.

Response 9: Indeed, our experiment can only indicate a potential effect on neurodegenerative diseases and cannot target Alzheimer's disease well. Therefore, we have made significant modifications to the article's theme and discussion. Thank you for your suggestion.

Comments 10: Overall, I suggest a lot more improvement is required before this manuscript goes for publication.

Response 10: Finally, once again, thank you for spending so much time helping us and giving us so many very valuable suggestions. We will do our best to improve this manuscript. We also hope that you can review the newly uploaded manuscript again and give us new opinions and insights.

Reviewer 4 Report

Comments and Suggestions for Authors

The study of proteins and low-molecular compounds of insects is very important for modern neurobiology and medicine. The authors conducted a fairly extensive study. However, I have doubts about the results of the experiments: insufficient repetition (even in experiments where large laboratory animals are killed in pharmacology, 6-fold repetition is standard; if animals are not killed, the minimum repetition is 6-8 times), incorrect statistical data processing (a correct method of multiple comparison of samples is needed, for example, the Tukey test), outdated methods of displaying results on diagrams (the mean ± standard deviation is not recommended for such small samples today, it is better to use the median, the first and third quartiles, the minimum and maximum values). In order for the results to be repeated in another laboratory, it is necessary to describe in great detail all the procedures during which some substances can be oxidized, that is, their pharmacological activity will be lost. The large volume of research and various methods are a positive side of the article, but the low quality of individual parts of the text and figures is a weak side of this article. The authors need to bring each part of the article to perfection, otherwise the article as a whole will raise more questions than answers.

1. Line 112: how many days were the insects stored in a dried state? At what temperature were they dried? What method of killing the insects was used? If the cockroaches were killed by freezing, then how long were they stored in a frozen state?

2. Line 113: "mass of the herb"? This text was probably copied from another article.

3. Lines 114-117: all procedures should be described in detail, since this is important for assessing the safety of substances oxidized by atmospheric oxygen at elevated temperatures.

4. Table 1: the units of measurement "minutes", "%" and the letter "B" do not need to be repeated in each cell.

5. Line 378: What is "x ± s": ± standard deviation or ± standard error? If the authors do not distinguish between these terms, the article cannot be accepted for publication. Tukey's test must be applied. Methods that incorrectly estimate differences between samples for non-normal samples (with reliable asymmetry or kurtosis) cannot be used.

6. Figure 4: I am not familiar with this method. What valuable information can the reader take away from this figure? The sizes of the numbers and letters do not allow reading anything in this figure.

7. Figure 5: The numbers are very small. Nothing can be read.

8. Line 485: The title of this figure (and other figures and tables as well) should be self-contained. That is, without reading the entire "Materials and Methods" section, the reader should be able to understand from the title of the figure what is being discussed and what method of comparing samples was used. In Figure 6, the data should be presented as a box analysis (median, first and third quartiles, minimum and maximum values). The divisions on the ordinate axis should be shown more accurately: minimum - 70, maximum - 120, division step - 5. A correct method of multiple sample comparison should be used, for example, the Tukey test.

9. The same remark applies to Figures 8, 10, 11, 12, 13, 15, 16, 17, 18.

10. Parts of the figures should be labeled a, b, c, e. For example, Figures 11, 12, 13.

11. What is the unit of measurement on the ordinate axis in Figure 12a?

12. Why is the repetition equal to 3 in Figures 10, 11, 15, 17, 18? Such repetition is not enough to formulate reliable conclusions.

13. Figure 7: in the lower corner, instead of “x100”, place a bar of 5, 10 or 50 micrometers in length. The size of the letters A, B, C, ... should be reduced by three times. In the title of the figure, describe the change in the shape of the cells in more detail.

14. The same remark applies to Figure 9.

15. In all figures, do not use bold font. Bold font is the equivalent of a raised voice in oral speech. Raising your voice is not accepted in science.

16. Line 525, 527, 533, 536 and others: if the authors use one of the thresholds for the reliability of differences in the diagrams, for example, 0.05, then in the text of the article, the specific value of the reliability of differences should be indicated in brackets, for example, P = 3.9 * 10-7.

17. Nothing can be read in Figure 13. This is disrespectful to the readers of the article. 18. Figure 14: it is enough to leave only the inscription "50 microns" on the lower right photograph. The title of the figure should be made 10 times longer: readers should read the title and see specific differences in each version of the photograph. All abbreviations should be deciphered: the figure should become self-sufficient.

19. Figure 18a is of very low quality. The image cannot be grouped into fragments. It should be a whole photograph.

20. The Discussion should be structured into 5-6 subsections. In this case, the authors will see weak subsections that require more detailed argumentation from literary sources. Dividing the text into subsections will lead to the removal of uninformative sentences (controversial, poorly supported by references to publications), which are saturated in the Discussion section.

21. Line 578, 653 and others: sentences should be formulated so that the names of the authors of the articles are absent from the discussion text.

22. Line 587 is a controversial statement. Give reasons.

23. Many sentences in the Discussion are unspecific, vague, and very generalized. The authors' thoughts flow from general formulations through methods to individual facts. And so on in a circle. Evidence-based medicine loves precision of formulations. It is necessary to discuss the results of your research, compare them with studies of similar depth and scope.

24. Line 700: what is this?

25. Journal titles should not be written in capital letters (e.g. lines 726, 727).

26. The generic and specific names in Latin should be italicized (e.g. lines 745, 747).

27. In the article title, "(L.)" needs to be removed.

Author Response

We are really very grateful to the reviewer for conducting such a careful and thorough review of our manuscript and giving us so many precious and highly valuable suggestions. This makes us realize that there are so many deficiencies in our manuscript writing, and at the same time, we feel ashamed of our poor English word choices. We have modified all the icons in the article, and re-analyzed the statistical methods using the Tukey analysis method. Moreover, we have changed all the statistical graphs from bar graphs to box plots for representation. Meanwhile, combining the opinions of other reviewers, we have changed the title and theme of the manuscript and made rectifications to the entire text. We hope that you can take a look at our re-uploaded manuscript and give us new comments to help us further improve the manuscript. Also, below is our reply to the rectification of the issues you raised previously. Please have a look:

Comments 1: Line 112: how many days were the insects stored in a dried state? At what temperature were they dried? What method of killing the insects was used? If the cockroaches were killed by freezing, then how long were they stored in a frozen state?

Response 1: At present in China, there are already three marketed drugs using Periplaneta americana as raw material herbs (Kangfuxin, Xinmailong, and Ganlong), and there are several standardized Periplaneta americana breeding factories. Our medicinal materials come from these standardized breeding factories.

Killing and drying methods: The breeding workshop is heated to 50-60°C for 4 hours for thermal killing. After killing, the insects are collected and washed, and then dried by hot air at 60-70°C. After drying, they are stored in a cold storage at 4-8°C and are generally provided to pharmaceutical factories within 2 months.

The extraction method of PAS-840: After the dried insects are crushed, they are soaked and extracted with 90% ethanol. The ethanol solution is concentrated under reduced pressure at 60°C and then degreased. The resulting degreased substance is chromatographed using S-8 macroporous adsorption resin. First, it is eluted with a certain amount of water, and then eluted with 40% ethanol. The ethanol eluate is concentrated under reduced pressure at 60°C to obtain the PAS840 sample. Here, PA represents Periplaneta americana, S8 represents the S-8 macroporous adsorption resin, and 40 represents elution with 40% ethanol.

Comments 2: Line 113: "mass of the herb"? This text was probably copied from another article.

Response 2: Thank you for your query. What we originally intended to express was adding ethanol at five times the volume of the drug. However, due to a translation error, the drug was wrongly translated as "herb." We have already made the modification at line 124 of the new manuscript, and changed the expression to adding 90% ethanol at five times the mass of the insect body.

Comments 3:Lines 114-117: all procedures should be described in detail, since this is important for assessing the safety of substances oxidized by atmospheric oxygen at elevated temperatures.

Response 3: Please refer to Comments 3 for this issue. We believe that detailing this process is not the focus of the article and will take up a large amount of space, so it has not been described in detail in the text.

Comments4: Table 1: the units of measurement "minutes", "%" and the letter "B" do not need to be repeated in each cell.

Response 4: Thank you for your query. We have already modified Table 1 of the manuscript, and there is a new version of the table at line 175 of the new manuscript.

Comments 5: Line 378: What is "x ± s": ± standard deviation or ± standard error? If the authors do not distinguish between these terms, the article cannot be accepted for publication. Tukey's test must be applied. Methods that incorrectly estimate differences between samples for non-normal samples (with reliable asymmetry or kurtosis) cannot be used.

Response 5: Thank you for the reviewer's suggestion. We have already re-analyzed all the data in the article using the Tukey analysis method and changed the "x ± s" below to "Mean ± median."

Comments 6:Figure 4: I am not familiar with this method. What valuable information can the reader take away from this figure? The sizes of the numbers and letters do not allow reading anything in this figure.

Response 6: This is a method for visualizing target-pathway-drug components in network pharmacology, which enables readers to directly observe which proteins and signal pathways have more connections with our drug components. It is a summary statement to prove that the previous three analysis modules have good relevance. Meanwhile, we also realized that the text in the picture was too blurry. We have enlarged the font size and adjusted the picture method, and we have also made rectifications for similar subsequent issues.

Comments 7: Figure 5: The numbers are very small. Nothing can be read.

Response 7: We are really sorry about the picture size. We have already made rectifications to the picture size.

Comments 8: Line 485: The title of this figure (and other figures and tables as well) should be self-contained. That is, without reading the entire "Materials and Methods" section, the reader should be able to understand from the title of the figure what is being discussed and what method of comparing samples was used. In Figure 6, the data should be presented as a box analysis (median, first and third quartiles, minimum and maximum values). The divisions on the ordinate axis should be shown more accurately: minimum - 70, maximum - 120, division step - 5. A correct method of multiple sample comparison should be used, for example, the Tukey test.

Response 8: Thank you for the reviewer's suggestion. We have already rectified all the analysis methods and presented the data using box plots that include the median, maximum, and minimum values instead of bar graphs.

Comments 9: The same remark applies to Figures 8, 10, 11, 12, 13, 15, 16, 17, 18.

Response 9: Thank you for your query. We have already carried out the corresponding rectifications.

Comments 10: Parts of the figures should be labeled a, b, c, e. For example, Figures 11, 12, 13.

Response 10: Thank you for your query. We have already carried out the corresponding rectifications. We have labeled all the composite pictures with a, b, c, and provided one-by-one annotations in the figure captions.

Comments 11: What is the unit of measurement on the ordinate axis in Figure 12a?

Response 11: Thank you for the reviewer's question. We have provided the full name annotation for it. This data is derived from the fluorescence values measured by a multifunctional microplate reader and does not have a specific unit name. The commonly used statistical expression is "fluorescence intensity."

Comments 12: Why is the repetition equal to 3 in Figures 10, 11, 15, 17, 18? Such repetition is not enough to formulate reliable conclusions.

Response 12: Thank you for the reviewer's query. Our experiment was conducted based on relatively stable in vitro experiments. In in vitro tests, we used cell lines with very small individual differences. Usually, three biological replicates are required, which means three batches of cell experiments completed at different time periods, and three technical replicates. Our n = 3 was carried out in accordance with the operational standards for in vitro experiments provided by our unit. However, your opinion, reviewer, is also very valuable for reference. We will complete the biological replicates with the standard of n = 6 - 8 in the subsequent in vivo animal experiments that we will continue to explore.

Comments 13:Figure 7: in the lower corner, instead of “x100”, place a bar of 5, 10 or 50 micrometers in length. The size of the letters A, B, C, ... should be reduced by three times. In the title of the figure, describe the change in the shape of the cells in more detail.

Response 13: Thank you for the reviewer's opinion. We have already made changes to all the cell photos by adding a clearly visible micrometers in length at the bottom right corner of the last picture in each group of photos. Also, we have rectified other similar issues.

Comments 14: The same remark applies to Figure 9.

Response 14: We have already made the corresponding rectifications.

Comments 15: In all figures, do not use bold font. Bold font is the equivalent of a raised voice in oral speech. Raising your voice is not accepted in science.

Response 15: Thank you for the reviewer's question. We have thoroughly checked the annotations of all the pictures and found that we did not use bold for annotation. It might be because the black font appears relatively prominent, giving the impression of being bold. We have reduced the font size of some fonts, and we hope that this rectification will satisfy you.

Comments 16: Line 525, 527, 533, 536 and others: if the authors use one of the thresholds for the reliability of differences in the diagrams, for example, 0.05, then in the text of the article, the specific value of the reliability of differences should be indicated in brackets, for example, P = 3.9 * 10-7.

Response 16: Thank you for the reviewer's question, after analyzing with SPSS, we found that for P - values with significant differences, specifically those less than 0.1, the complete scientific-notation numbers cannot be fully displayed.

Comments 17: Nothing can be read in Figure 13. This is disrespectful to the readers of the article.

Response 17: Thank you for your criticism. We have already rectified the pictures. Through picture adjustment, Figure 13 has now been renamed Figure 10, and we have enlarged it and added the statistical chart that was previously placed in Figure 14.

Comments 18: Figure 14: it is enough to leave only the inscription "50 microns" on the lower right photograph. The title of the figure should be made 10 times longer: readers should read the title and see specific differences in each version of the photograph. All abbreviations should be deciphered: the figure should become self-sufficient.

Response 18: We have already rectified the picture, and now Figure 14 has officially been renamed Figure 11.

Comments 19: Figure 18a is of very low quality. The image cannot be grouped into fragments. It should be a whole photograph.

Response 19: Thank you for your suggestion. However, for the original Figure 18 (which is now Figure 14), we received an email from the editor before peer review, and at the request of the journal editor, we integrated the WB dot plot. This is the reason why it was highlighted in yellow at the beginning of the manuscript. But we have also made some rectifications. We have re-analyzed its bar graph and replaced it with a box plot.

Comments 20: The Discussion should be structured into 5-6 subsections. In this case, the authors will see weak subsections that require more detailed argumentation from literary sources. Dividing the text into subsections will lead to the removal of uninformative sentences (controversial, poorly supported by references to publications), which are saturated in the Discussion section.

Response 20: Thank you very much for your suggestion, reviewer. We deeply recognize our weakness in language proficiency. We have rewritten the discussion section, and we also hope that the reviewer can provide new comments on it to help us continue to improve this manuscript.

Comments 21: Line 578, 653 and others: sentences should be formulated so that the names of the authors of the articles are absent from the discussion text.

Response 21: Thank you for your suggestion. We have already modified similar expressions.

Comments 22,23: Line 587 is a controversial statement. Give reasons,Many sentences in the Discussion are unspecific, vague, and very generalized. The authors' thoughts flow from general formulations through methods to individual facts. And so on in a circle. Evidence-based medicine loves precision of formulations. It is necessary to discuss the results of your research, compare them with studies of similar depth and scope.

Response 22,23: We have already reorganized and rewritten the discussion section.

Comments 24: Line 700: what is this?

Response 24: Sorry, this is an example sentence included in the manuscript template provided by the MDPI company, and we neglected to delete it when transferring the manuscript content.

Comments 25, 26: Journal titles should not be written in capital letters (e.g. lines 726, 727). Journal titles should not be written in capital letters (e.g. lines 726, 727).

Response 25, 26: Sorry, we also realize that abbreviations should not be used in the journal title. Meanwhile, we have modified the journal name by capitalizing the first letter. We have already rectified the references.

Comments 27: In the article title, "(L.)" needs to be removed.

Response 27: Thank you for your opinion. We have already rectified the title.

Round 2

Reviewer 1 Report

Comments and Suggestions for Authors

I would like to thank the authors of the manuscript entitled Periplaneta americana (L.) extract alleviates the disease progression of Alzheimer's disease by inhibiting the expression of NMDAR in nerve cells” for promptly addressing the requests and comments made during the review process. Furthermore, I commend them for their meticulous attention to corrections in the use of the English language, which has ensured a clear and high-quality presentation of the work.

Author Response

Thank you very much for your support and suggestions. We also realize that there are still many deficiencies in our language usage. If necessary, we are also willing to seek language polishing services from the MDPI company to help our manuscript better conform to the writing style of scientific manuscripts in terms of language.

Reviewer 2 Report

Comments and Suggestions for Authors

Here are some suggestion below:

·       In result section, NMDA receptor western is not very clear, author should add better image for more clarity.

·       In summary option, author don’t need add protein details, it should be simple for readers.

·       Abstract should be rewrite. Even all data and content author has added but author should use impressive and scientific words.

·       Overall manuscript is well designed and adding insight in future research. But author should correct the English language.

Comments on the Quality of English Language

need to improve the language.

Author Response

Thank you, reviewer, for your prompt reply. We have revised the manuscript in accordance with your suggestions and have provided corresponding responses to your queries. Should you have any additional questions or suggestions regarding our manuscript, we will be actively engaged in responding and further revising it. We are truly grateful for your assistance in helping us improve the manuscript to an even greater extent.

Comments 1: In result section, NMDA receptor western is not very clear, author should add better image for more clarity.

Response 1: Thank you for pointing out our deficiencies. We have already modified Figure 14.

Comments 2: In summary option, author don’t need add protein details, it should be simple for readers.

Response 2: Thank you for your suggestions. We have already streamlined and revised the content of this part.

Comments 3: Abstract should be rewrite. Even all data and content author has added but author should use impressive and scientific words.

Response 3: Thank you very much for your suggestions. I am aware that our language proficiency is seriously inadequate. We have revised the content of the abstract to the best of our ability. Meanwhile, if necessary, we will also seek the language editing service of the MDPI company to help us perfect the manuscript.

Reviewer 3 Report

Comments and Suggestions for Authors

The manuscript has been significantly improved. I have pretty minor comments:

1. Please explain the significance of using PC12 cells.

2. Are there any plans for in vivo investigations or clinical trials to confirm PAS840's neuroprotective potential?

3. The manuscript discusses anti-inflammatory and antioxidant properties. Were certain pathways, such as NF-kB and ROS scavenging, definitively identified as mediators?

4. Is there a risk of overstating PAS840's medicinal potential based on in vitro results?

5. To what extent can the findings be applied to models of excitotoxicity or other neurodegenerative diseases beyond Alzheimer's disease?

6.  Gaps in knowledge that need to be addressed, such as long-term efficacy or pharmacokinetics of PAS840?

Comments on the Quality of English Language

It has been improved compared to the previous version of the manuscript.

Author Response

Thank you very much for your recognition. We also sincerely hope to obtain more of your insights and suggestions to help us continue to improve our manuscript. Meanwhile, we have also provided corresponding responses to your questions.

Comments 1: Please explain the significance of using PC12 cells.

Response 1: The PC12 cell line is derived from a rat adrenal pheochromocytoma monoclonal transplanted to the adrenal gland, and it exhibits neuronal-like functional characteristics after differentiation, and is widely used in in vitro neurobiology and neurological disease research. In this study, we used rat-derived PC12 cells instead of human-derived SH-SY5Y cells. This was because we happened to have the PC12 cell line in the research group at that time, and we did not give it too much careful consideration. This was our oversight. However, as a cell similar to nerve cells, the effectiveness of drug protection on PC12 cells can, to some extent, provide support for the study of the protective effect of drugs on the nervous system at the animal level.

Comments 2: Are there any plans for in vivo investigations or clinical trials to confirm PAS840's neuroprotective potential?

Response 2: At present, we are preparing to conduct an Alzheimer's disease (AD) animal model experiment using APP/PS1 double transgenic mice as the test subjects with PAS840. It is proposed that at the time of drug administration for 3 months (6 months old), 5 months (8 months old), and 7 months (9 months old), the Morris experiment will be used to test the changes in the cognitive impairment process of mice in each group. And at three time points, namely before drug administration, after 3 months of drug administration, and after 7 months of drug administration, the degree of Aβ deposition and Tau protein phosphorylation in the mouse brain tissue will be detected and compared through brain pathological section staining and Western blotting (WB), so as to verify the effectiveness of PAS840 in improving the AD process of mice. Meanwhile, at the above-mentioned time points, multi-faceted and multi-angular test analyses will be carried out on the brain tissue transcriptome, brain tissue metabolome, serum metabolome, intestinal microbiota, and microbial metabolites of mice in each group. Clinical research is valuable only after a large number of animal experiments have been precisely verified, and there are currently no plans for it.

Comments 3: The manuscript discusses anti-inflammatory and antioxidant properties. Were certain pathways, such as NF-kB and ROS scavenging, definitively identified as mediators?

Response 3: Due to the limited funding in the early stage, the graduation time of our first author is approaching, and this manuscript needs to be published to meet the graduation requirements, so we consider this manuscript as the preliminary basic research of this project. We have not conducted an in-depth study on its core mechanism. However, when designing in vivo experiments, we will screen out the most core differential genes by comparing AD mice, normal mice, and drug-administered mice based on the macroscopic data of proteomics and transcriptomics, so as to analyze the core therapeutic mechanism of the drug on the disease.

Comments 4: Is there a risk of overstating PAS840's medicinal potential based on in vitro results?

Response 4: This indeed poses a risk of exaggerating the pharmaceutical potential. After all, the research has only obtained effective results at the in vitro cell experiment level, and further evidence of its effectiveness is needed at the animal level. In the later stage, we will conduct relevant research work through in vivo experiments to confirm its effectiveness.

Comments 5: To what extent can the findings be applied to models of excitotoxicity or other neurodegenerative diseases beyond Alzheimer's disease?

Response 5: This issue is still unpredictable because we have not yet carried out the above type of model study at the in vivo experiment level. However, we have conducted a study on the protective effect of this PAS-840 sample on stroke in rats (Investigation of the effects of Periplaneta americana (L.) extract on ischemic stroke based on combined multi-omics of gut microbiota DOI: 10.3389/fphar.2024.1429960). It does help with stroke and recovery after stroke in rats, indicating that it has a certain protective and repair effect on brain nerve damage caused by ischemia.

Comments 6: Gaps in knowledge that need to be addressed, such as long-term efficacy or pharmacokinetics of PAS840?

Response 6: There are indeed many problems that need to be solved in the research. The main problem that our team currently aims to address is to first verify the effectiveness of PAS840 in vitro experiments, and then conduct in vivo experiments to detect the pharmacokinetic indices of PAS840, and further explore whether long-term administration will lead to toxic side effects and reduced drug effectiveness. Through the ischemic stroke experiment mentioned in the previous question, we found that drug administration can effectively enhance the neuronal activity of rats and improve their resistance to sudden nerve damage during long-term administration. Therefore, we are considering whether PAS840 can have protective and therapeutic effects on various neurological diseases. This also requires us to explore from more perspectives the pharmacological effects that this drug itself can produce and its impact on the internal microenvironment, thereby improving the course of neurodegenerative diseases. As a result, we are preparing to conduct relevant research through in vivo experiments and resolve the aforementioned issues after confirming its drug efficacy.

Reviewer 4 Report

Comments and Suggestions for Authors

This article has been sufficiently improved by the authors and can be recommended for publication.

Author Response

Thank you ever so much for your wonderful support! We are truly delighted to have your input and would very much appreciate it if you could continue reviewing our manuscript. If you happen to have any suggestions or ideas on how we can improve our manuscript, please do let us know. We'll put in our utmost effort to make this manuscript even better. Once again, we want to express our heartfelt gratitude for your meticulous review and your incredibly helpful suggestions. Your time and effort are invaluable to us.